# Integrative pan-resistome and transcriptomic characterization reveals differential gene expression signatures in carbapenem-resistant *Acinetobacter baumannii*: Insights into efflux pump regulation and therapeutic targeting strategies

Mohanraj Gopikrishnan, George Priya Doss C*

Department of Integrative Biology, School of Biosciences and Technology, Vellore Institute of Technology (VIT), Vellore, Tamil Nadu, India

* georgepriyadoss@vit.ac.in

## Abstract

*Acinetobacter baumannii* is a significant, multidrug-resistant pathogen that is increasingly recognized as an agent associated with hospital infections. Its increasing resistance to carbapenems and other necessary antibiotics poses a serious threat to public health worldwide. The present study provides an integrative analysis of the pan-resistome and transcriptomic landscape of carbapenem-resistant *A. baumannii* (CRAB) under sub-minimum inhibitory concentrations (sub-MICs) of clinically relevant antibiotics, i.e., ciprofloxacin, amikacin sulfate, meropenem, and polymyxin B. A focused investigation was conducted into the transcriptional modulation of efflux transport systems and cellular stress-response mechanisms. To identify differentially expressed genes (DEGs) among the CRAB strains, parallel comparative RNA-Seq analysis with already available public datasets was undertaken. Concurrently, high-throughput virtual screening against the comprehensive marine natural product database (CMNPD) was done to identify inhibitors for MacB, a major ABC-type efflux transporter. Binding stability and interaction profiles of lead compounds were assessed via molecular dynamics simulations of 1000 ns. Transcriptomic profiling consistently showed increased levels of MacA-MacB efflux components, RcnB an intracellular stress-response protein LolA an outer membrane chaperone and surface antigen protein 1, especially under polymyxin B exposure. CMNPD27284 is the best candidate, having a strong binding affinity and stable interaction networks with critical MacB residues (−7.20 kcal/mol). These results highlight that efflux-mediated resistance and stress adaptation are crucial factors in CRAB, and they also indicate CMNPD27284 as a potential candidate for marine-derived scaffolding in developing drugs targeting the efflux pump.

**Data availability statement:** All relevant data are within the paper and its Supporting Information files.

**Funding:** The author(s) received no specific funding for this work.

**Competing interests:** The authors have declared that no competing interests exist.

## 1. Introduction

The global proliferation of carbapenem-resistant *Acinetobacter baumannii* (CRAB) presents a formidable challenge to contemporary healthcare systems, particularly within nosocomial environments, due to its extensive multidrug resistance profile [1]. In addressing this growing threat, a paradigm shift is required from single-genome to pangenomic analyses that considers the broad spectrum of genetic diversity across the species [2]. The pangenome is the entire gene collection of a bacterial species, consisting of core genes that all strains have and are essential for survival, and accessory genes that differ in strains and confer particular attributes [3]. This architecture creates a suitable environment for investigation into mechanisms of adaptation, virulence factors, and determinants of antimicrobial resistance [4]. More importantly, *A. baumannii* possesses a big pangenome, or it remains "open" as it continues to expand with the sequencing of new isolates, attesting to its remarkable genomic plasticity and thus, its propensity for horizontal gene transfer (HGT). Such a highly dynamic genomic landscape permits the rapid acquisition of either resistance genes, mobile genetic elements, or pathogenicity islands, thereby maximizing survival and persistence in clinical settings [5–7].

Pangenomic analysis enables the comparative resistome profile across different isolates [8]. It thus reflects the evolutionary trajectories and distribution patterns of various resistance determinants, such as β-lactamases (OXA and NDM types), efflux systems (AdeABC, MacAB-TolC), aminoglycoside-modifying enzymes, and insertion sequences [9–11]. Conjunctively, conserved resistance genes can be recognized as universal therapeutic targets, while accessory genes often underlie isolate-specific or extreme resistance phenotypes. Thus, pangenomics is an ideal path to decode the evolution of resistance and set priorities in drug development targets [12,13]. However, the presence of resistance genes does not guarantee functional expression under antimicrobial stress. To bridge this gap, transcriptomic profiling, particularly RNA-Seq, throws a dynamic light on gene expression and unveils regulatory networks governing bacterial adaptation to antibiotic pressure [14]. Genes exhibiting transcriptional upregulation under stress conditions are likely contributors to survival pathways [15]. Investigating gene expression under sub-minimum inhibitory concentrations (sub-MICs) of antibiotics provides a physiologically relevant model, mimicking conditions encountered during biofilm formation or suboptimal treatment. Under such stress, CRAB activates adaptive responses including efflux pump overexpression, membrane remodeling, oxidative stress mitigation, and metabolic reprogramming [16,17]. Of particular significance, the MacA–MacB efflux complex demonstrated marked transcriptional induction under antibiotic exposure, attributed to its broad-spectrum substrate extrusion capabilities [18].

A systems biology framework integrating pangenomic and transcriptomic data offers a powerful approach to unravel AMR mechanisms within microbial ecosystems. While the pangenome covers the entire genetic potential, the transcriptome covers the actively expressed subgenes under current selective pressure [19]. Such an integration allows the identification of conserved-transcriptionally active resistance genes

with great therapeutic potential, especially for more complex pathogens like *A. baumannii*, wherein resistance arises more from intertwined genetic networks than through single mutations. In this study, pan-resistome mapping and differential transcriptomic analysis were conducted on CRAB exposed to sub-MIC levels of four clinically significant antibiotics: ciprofloxacin, amikacin sulfate, meropenem, and polymyxin B. The genes that were consistently upregulated after the treatment were considered as resistance drivers. Among these, the MacB efflux transporter was selected as a potential target owing to its high substrate variety and significant transcriptional upregulation [20]. High-throughput virtual screening of the Comprehensive Marine Natural Product Database (CMNPD) was undertaken to identify potential inhibitors, and molecular dynamics simulations (MDS) were used to validate ligand-target interactions and binding stability. CMNPD27284 showed favorable binding energetics and binding stability, highlighting the promise of using marine-derived scaffolds as efflux pump inhibitors. Altogether, this open strategy, capable of pangenome-transcriptome-drug discovery, provides an umbrella framework for deciphering the mechanism of resistance in CRAB. This approach allows genomic variation related to transcriptional activation to be understood, and ideally, new inhibitors can be validated through computational modeling. The entire methodological workflow is shown in Fig 1.

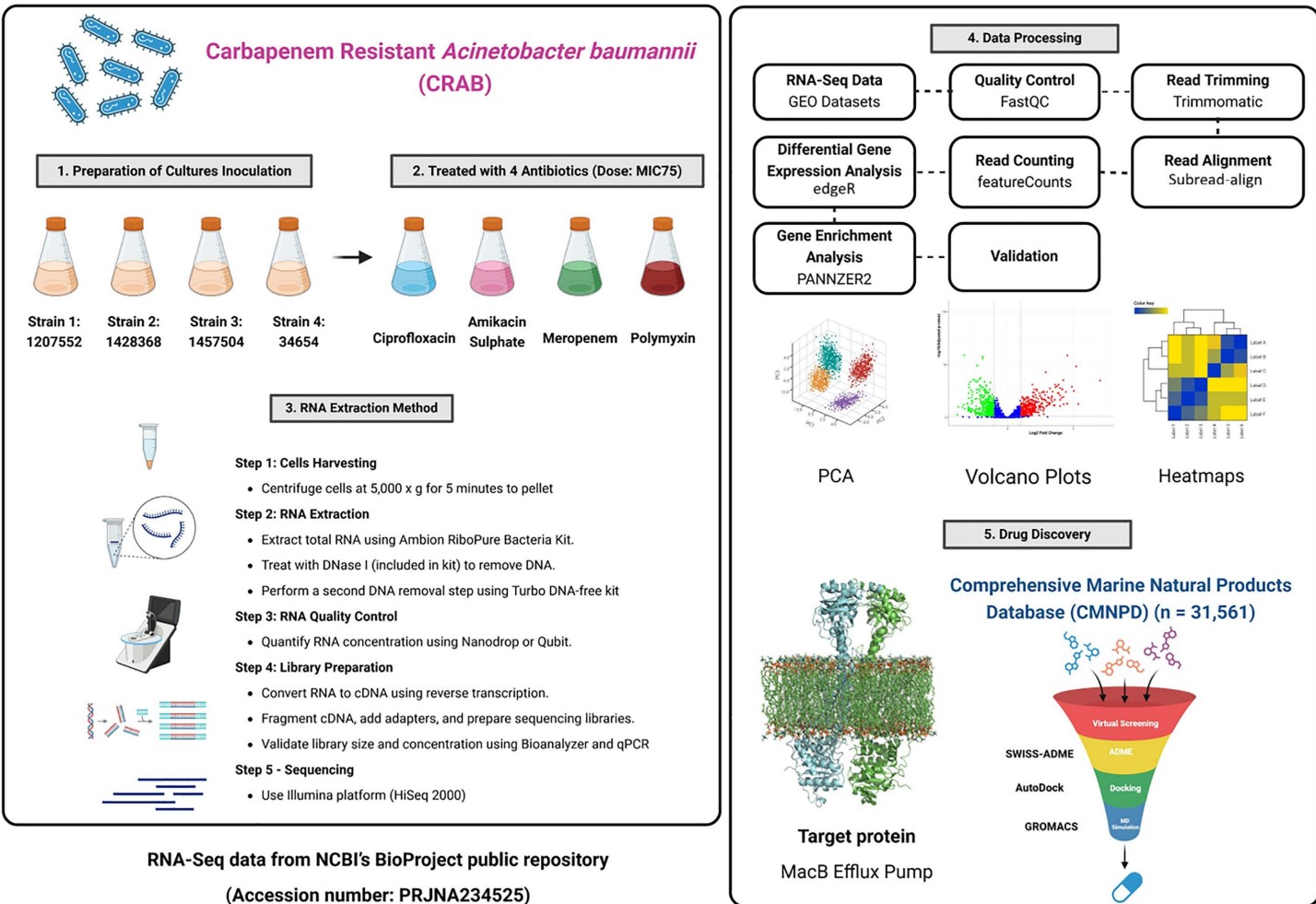

**Fig 1. Schematic outline of the computational pipeline in this study combining transcriptomic profiling, differential gene expression analysis, virtual screening of the marine-derived bioactives, and molecular docking based on conserved residues in the MacB efflux transporter.**

## 2. Materials and methods

### 2.1. RNA-Seq data acquisition and pre-processing

Transcriptomic datasets for *A. baumannii* strains relevant to the closure of the study were obtained from Gene Expression Omnibus (GEO) [21], with accession numbers shown in Table 1. The raw sequencing reads were retrieved from the Sequence Read Archive of the NCBI [22] and were processed into FASTQ format for further bioinformatics analyses. Draft genome assemblies were obtained from the NCBI Assembly database [23] to serve as strain-specific references for transcript alignment. Initial quality control of raw reads was performed using FastQC (v0.11.9), followed by adapter trimming and quality filtering with Trimmomatic (v0.39), applying a sliding window of 4:20 and a minimum read length threshold of 36 bp [24]. With the help of Subread-align (v2.0.3), all good reads were aligned to reference genomes using this high-performance aligner tailored for RNA-Seq data [25]. Quantification for every gene was performed with featureCounts (v2.0.3), allowing only certain reads that had been uniquely mapped [26]. Gene annotations were obtained from NCBI RefSeq release 220 in GTF format to maintain compatibility with quantification tools [27]. A total of 32 RNA-Seq libraries were analyzed for four *A. baumannii* strains subjected to duplicate treatments with ciprofloxacin, amikacin, meropenem, and polymyxin B. The count matrices produced from this were used in differential expression analysis with edgeR.

### 2.2. Strain typing, pan-genome construction, and visualization

The PubMLST platform, which uses a species-specific Multi-Locus Sequence Typing (MLST) scheme to draft genomes into allelic profiles and assign STs for the indicated organism in order, was used for strain typing [28,29]. The clinical isolates were GCA_000584495.1 and GCA_000588575.2 (sputum-derived), and GCA_000581515.1 and GCA_000584375.2 (perirectal-derived). All strains were taxonomically confirmed as *A. baumannii* using a 16S rRNA BLAST analysis and were also typed as ST2, corresponding to the International Clone II. Selection criteria also included clinical site diversity as well as sequence type. Thereafter, the genome assemblies were downloaded from NCBI and evaluated for quality using N50 values and completeness (>95% for all strains). All four clinical isolates were then put through pan-genomic analysis together with their reference strain ATCC 17978 (GCA_000745985.1) through the Roary pipeline [30]. Roary's used to identify core and accessory genes across the genomes analyzed. In this case, the core genes are defined as the ones with >95% amino acid identity and present in more than 99% of genomes included in the study. Additional parameters have included a threshold of 95% BLASTp identity and a core gene inclusion cutoff of 99%. The resulting presence-absence matrix was visualized using Phandango (an interactive tool that integrates gene content variation with phylogenetic context) [31]. The Roary-derived matrix and phylogenetic tree were co-uploaded into Phandango, enabling dynamic visualization of conserved and strain-specific genes. Gene clusters represented by different colors would provide comparative insight into core and accessory gene architecture from a genomic configuration perspective.

### 2.3. Antimicrobial resistance and virulence gene prediction

Antimicrobial resistance (AMR) and virulence gene profiling via Abricate (https://github.com/tseemann/abricate) [32]. Draft assemblies screened against the Comprehensive Antibiotic Resistance Database (CARD v4.0.1) [33] and VirulenceFinder

**Table 1.** Represents the RNA-Seq datasets used, including antibiotics at 0.75×MIC, biological replicates, species confirmation through 16S rRNA BLAST and MLST, and some key metrics from annotation.

| Accession | Strain | Antibiotic Treatment (0.75×MIC) | Verification (16S rRNA BLAST) | Source | MLST Sequence Type |
|---|---|---|---|---|---|
| GSE56218 | 34654 | Ciprofloxacin, Amikacin, Meropenem, Polymyxin B | *Acinetobacter baumannii* | Perirectal | ST2, International Clone II |
| GSE56222 | 1207552 | Ciprofloxacin, Amikacin, Meropenem, Polymyxin B | *Acinetobacter baumannii* | Sputum | ST2, International Clone II |
| GSE56223 | 1428368 | Ciprofloxacin, Amikacin, Meropenem, Polymyxin B | *Acinetobacter baumannii* | Sputum | ST2, International Clone II |
| GSE56224 | 1457504 | Ciprofloxacin, Amikacin, Meropenem, Polymyxin B | *Acinetobacter baumannii* | Perirectal | ST2, International Clone II |

(v6.0) [34]. Defining presence according to nucleotide identity ≥90% and coverage ≥80% and applying default parameters in Abricate unless specified otherwise. The same four strains of *A. baumannii* GCA_000584495.1, GCA_000588575.2 (sputum), and GCA_000581515.1, GCA_000584375.2 (perirectal) were analyzed. Taxonomic confirmation through 16S rRNA BLAST and MLST classification against the standards reaffirmed all strains as ST2, International Clone II. Thus, each strain's resistome and virulome have been holistically characterized by applying the approaches above to understand transcriptomic responses under antibiotic pressure.

## 2.4. Differential gene expression (DEGs) analysis

DEGs analysis was performed by edgeR (version 3.6.2; Bioconductor, R) [35]. The raw counts were normalized by trimmean of M-values (TMM) to take care of library size and composition biases. Statistical inference was conducted using the quasi-likelihood framework of edgeR, employing F-tests to contrast treated and untreated conditions. Two biological replicates were used to support the actual trials, providing some statistical robustness. DEGs were taken into consideration when the absolute log$_2$ fold change ($|log_2FC|$) > 1 and the FDR-adjusted p-value <0.01 (Benjamini Hochberg correction) was present. Using PCA to evaluate batch effects revealed batches with less overall variance. Visualization of DEGs was carried out in the ComplexHeatmap package in R [36], while shared DEGs across treatments and strains were shown as part of the results described in the VennDiagram package [37].

## 2.5. Functional enrichment and gene ontology (GO) analysis

ClusterProfiler R package [38] (version 4.10.1) was employed for GO enrichment analysis to investigate the biological functions associated with the DEGs. The DEGs were mapped to Entrez Gene IDs, and pertinent GO search terms were cross-referenced with the GO database to obtain a list of terms that are significantly enriched in Biological Processes (BP), Cellular Components (CC), and Molecular Functions (MF). The enrichment of GO terms was subjected to hypergeometric testing with FDR correction, and a cut-off p-value of < 0.05 was applied. The results are depicted in bar plots and enrichment maps, highlighting the significant biological pathways that have been altered upon antibiotic exposure.

## 2.6. Homology modeling and structural validation

The amino acid sequence of the MacB efflux transporter was retrieved from the NCBI protein database and subjected to homology modeling via the SWISS-MODEL server [39]. Based on the alignment and structural homology, the server selected the crystal structure with PDB ID 5WS4 [20] as the modeling scaffold. The PROCHECK module of the PDBsum online platform was used to assess the anticipated tertiary model's stereochemical quality [40]. Validation metrics included the Ramachandran plot, which graphically represents the φ (phi) and ψ (psi) backbone dihedral angles for all non-glycine, non-terminal residues. Glycine and terminal residues were analyzed separately due to their intrinsic conformational variability [41]. Model quality was considered acceptable for more than 90% of residues occupying the most favored regions of the Ramachandran space, indicating that the model is stable and stereochemically reliable.

## 2.7. Target structure optimization

Post-validation, the modeled MacB structure was refined for molecular docking applications. Preprocessing steps included energy minimization through the addition of hydrogen atoms, removal of water molecules in the crystal, and other nonessential heteroatoms from the structure. Optimization and visualization have been performed using PyMol [42], which was also used to identify putative pockets for ligand binding. Eventually, these active sites were used to set the docking grid for virtual screening.

## 2.8. Ligand library preparation

Ligands were sourced from the Comprehensive Marine Natural Products Database (CMNPD) [43], encompassing bioactive marine-derived phytochemicals. The antibiotic reference controls were Amikacin Sulfate (PubChem ID: 38351), Ciprofloxacin (PubChem ID: 2764), Polymyxin B (DrugBank ID: DB00781), and Meropenem (PubChem ID: 441130). All compounds were initially obtained in PDB format and converted to PDBQT format using AutoDock Tools to make them compatible with docking simulations. Ligands from the CMNPD were virtually screened with the MacB model, and those with better binding affinities than those of control antibiotics were selected for further pharmacological assessment.

## 2.9. Virtual screening and molecular docking

Molecular docking was conducted using AutoDock Vina [44], which predicts the ligand-receptor binding conformations and free energy scores associated with them. The receptor and ligand files were prepared using AutoDock Tools (ADT), which consists of a grid box configuration centered on the predicted active site. The docking outputs, including binding energies and spatial conformations, were analyzed using Discovery Studio 3.5 [45], which provides a detailed rendering of 2D and 3D interaction mapping of ligands with the MacB binding pocket.

## 2.10. *In Silico* ADMET profiling

The pharmacokinetics and drug-likeness properties of all top-performing ligands were studied in silico by ADMET analysis (SwissADME calculated ADMET properties) [46]. The canonical representation of the SMILES string for each compound was extracted from PubChem and put into the SwissADME to assess absorption, distribution, metabolism, excretion, and toxicity parameters. Such a computational screening yielded information about the bioavailability and therapeutic promise of candidate molecules, directing the choice of a lead compound for validation through experiments.

## 2.11. Molecular dynamics simulation (MDS)

To explore the conformational dynamics and ligand-mediated structural modulation of the MacB efflux transporter, MDS was carried out in both the unbound (apo) and ligand-bound (holo) states. System assembly and membrane embedding were performed using the CHARMM-GUI Membrane Builder [47], enabling the construction of a biologically relevant lipid bilayer environment composed of 1-palmitoyl-2-oleoyl-sn-glycero-3-phosphocholine (POPC). In both systems, the apo and holo proteins were placed in the bilayer portion of the membrane. The membrane was explicitly solvated with water and checked for neutrality using 0.15 M NaCl, placed with the distance method. Force field parameters had been assigned to the protein and lipid components using the CHARMM36m force field. Ligand topologies were produced using the General CHARMM force field via ParamChem [48]. Accurate membrane-protein orientation was ensured using the PPM server, aligning the protein such that acidic residues faced the aqueous phase and basic residues (lysines) were oriented toward the membrane interface. The simulation box was defined as a rectangular prism (80 Å × 80 Å × variable Z) with a 22.5 Å water layer on each side. The bilayer comprised 72 POPC molecules in the upper leaflet and 67 in the lower leaflet. Following energy minimization, the system underwent thermal equilibration to 300 K over 1 ns under an NVT ensemble. A 5 ns NVT equilibration phase with positional restraints on the protein backbone and progressive relaxation of lipid constraints succeeded this. Final equilibration was performed under NPT conditions using the Parrinello–Rahman barostat and V-rescale thermostat to maintain 310 K and 1 bar pressure [49]. Using the Particle Mesh Ewald (PME) [50] approach for long-range electrostatics, the PME order will be 4. The spacing in the Fourier grid is 0.16 nm. All covalent bonds were constrained by applying the LINCS algorithm, and a 2-fs integration time step was employed for computations. Production simulations were performed for 1000 ns for both apo and holo systems using GROMACS [51].

## 2.12. Essential dynamics

Essential dynamic (ED) analysis for MacB was conducted to decipher the principal conformational transitions. The Cα positional fluctuations were used to form a covariance matrix using the GROMACS g_covar and g_anaeig utilities for their analysis. The main collective motion (eigenvector 1, EV1) was extracted and visualized by using PyMOL [52]. This was done to compare the dynamic behaviour of both the apo and holo states. To further characterize the conformational landscape, free energy landscape (FEL) [53] analysis was conducted with two reaction coordinates: structural deviation (for example, RMSD) and radius of gyration (Rg). The Gibbs free energy surface was then calculated from the appropriate Boltzmann relation:

$$\Delta G\ (p1,\ p2) = -k_B\ T\ \ln\rho(p1,\ p2)$$

Thus, the definitions are as follows: $\Delta G$ = free energy, $K_B$ = Boltzmann constant, T = the simulation temperature, and $\rho(p1,p2)$ = joint probability distribution over the involved reaction coordinates p1 and p2. Two-dimensional FEL maps have captured energetically favoured conformations and transition pathways. The Dynamic Cross-Correlation Matrix (DCCM) analysis was included to examine the correlations in movement between residues concerning ligand-induced motions. Variances of displacements were used to compute the cross-correlation coefficients ($C_{ij}$) using Bio3D in R [54].

$$C_{ij} = \langle \Delta r_i \cdot \Delta r_j \rangle / \{\langle \Delta r_i \rangle 2 \langle \Delta r_i \rangle 2\}^{1/2}$$

Here, $\Delta r_i$ and $\Delta r_j$ mean the deviation of atoms $i$ and $j$ from their mean positions; angular brackets indicate time-averaging. Thus, positive values of $C_{ij}$ imply correlated motions (parallel displacement), and negative values suggest anticorrelated dynamics (opposing displacement) with reference to allosteric communication and structural coupling in the protein.

## 2.13. MM-PBSA analysis

MM-PBSA calculations were employed to arrive at a quantitative estimate of the binding affinity between MacB and its high-affinity ligand [55]. The binding free energy ($\Delta G$ binding) was calculated using the 1000th frame of the production trajectory through the following thermodynamic cycle:

$$\Delta G\ (Binding) = G\ (Complex) - G\ (Receptor) - G\ (Ligand)$$

In this equation, G Complex denotes the free energy of the protein-ligand system at a given instance; G Receptor and G Ligand indicate the free energy of the isolated protein and ligand in the solvent, respectively. Using the g_mmpbsa tool within the GROMACS suite [56], binding energy could be decomposed into Van der Waals, electrostatic, polar solvation, and non-polar solvation components. This initial analysis provides a quantitative framework for discussing ligand binding stability and vigor, thus allowing the selection of possible efflux pump inhibitors.

## 3. Results

### 3.1. Transcriptomic dataset acquisition and strain verification

RNA-Seq datasets obtained from the GEO comprised *A. baumannii* strains subjected to sub-inhibitory concentrations (0.75 × MIC) of ciprofloxacin, amikacin, meropenem, and polymyxin B for 15 minutes, as summarized in Table 1. Each of these antibiotic treatments was done in biological duplicates per strain for reproducibility of the experimental data. Raw sequencing reads were obtained from NCBI SRA, checked for quality through FastQC, and trimmed with Trimmomatic to remove low-quality bases and adapter sequences. Finally, all trimmed reads were aligned against their draft genome

using the Subread aligner. Strain identities were confirmed by querying sequence data against NCBI genome assemblies to ensure accurate transcriptomic analysis at the strain-specific level (Supplementary File 1, Sheets 1–2). Notably, untreated controls were not included; differential expression analyses were conducted by comparing transcriptional responses across distinct antibiotic exposures.

### 3.2. Species confirmation and sequence typing (MLST)

Four clinical isolates were used for transcriptomic analysis: GSE56218, GSE56222, GSE56223, and GSE56224, which have GEO accession numbers. All samples were confirmed as *A. baumannii* with taxonomic identification using 16S rRNA BLAST analysis. The anatomical sites from which these isolates were recovered were sputum and perirectal swabs. Sequence typing was carried out using the Pasteur MLST scheme via the PubMLST database, which identified all four isolates to be Sequence Type 2 (ST2), belonging to the International Clone II (IC-II), a globally prevalent multidrug-resistant lineage (Table 1).

### 3.3. Pangenome composition and gene distribution

The comparative pan-genomic analysis of *A. baumannii* strains revealed a total of 5,222 non-redundant genes. Among them, 3,093 genes (59.2%) were identified as core genes frequently present across all isolates, while 2,129 genes (40.8%) were assigned shell gene status, indicating interstrain genomic variations (Fig 2). The absence of soft-core and cloud genes suggests that the genomic framework is conserved, with moderate levels of diversity in accessory genes. To see if the clinical source imposes a selective pressure on genomic composition, we have annotated the isolates by source (sputum vs. perirectal). A gene presence-absence matrix along with phylogenetic reconstruction showed a source-specific clustering: perirectal isolates (GCA_000581515.1 and GCA_000584375.2) formed one clade, whereas sputum-derived strains (GCA_000584495.1 and GCA_000588575.2) formed another. This observation suggests that selective pressures acting on the niche may be responsible for the genomic divergence that could, in turn, affect strain-specific adaptations and functional capabilities.

### 3.4. Resistome and virulome profiling

Analysis with ABRicate against the CARD database identified multiple antimicrobial resistance (AMR) genes across the strains. The most commonly detected resistance genes included *OXA-23-like, OXA-51-like, mph(E), msr(E), sul1, and sul2*, as shown in Fig 3A. Among the strains, strain 1457504 showed the broadest range of resistance determinants, with those for other aminoglycoside and macrolide resistance genes, indicative of a multidrug-resistant (MDR) phenotype. In

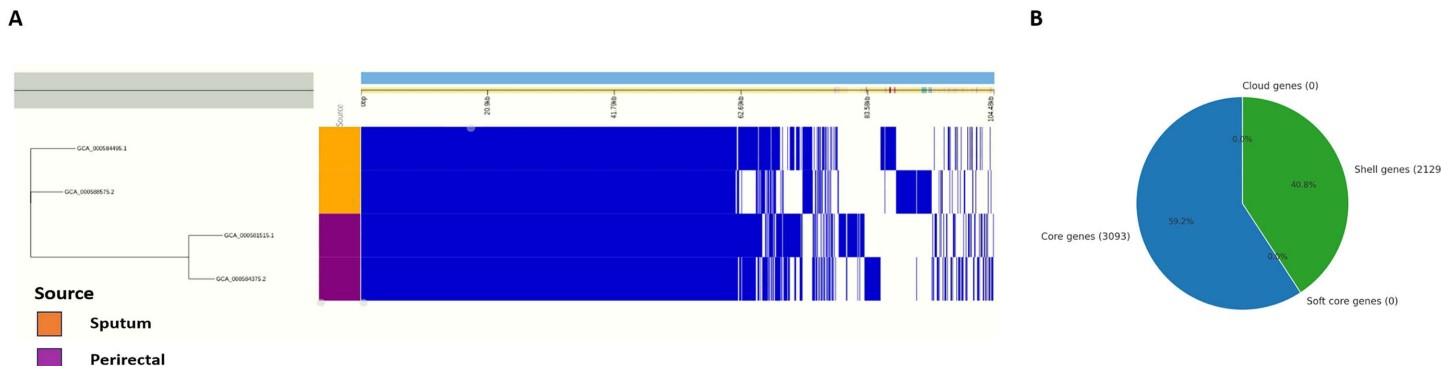

**Fig 2. Comparative pangenomic analysis of the four strains of *A. baumannii*. (A)** The Phandango Visualization describes presence (blue) and absence (white) states of genes across core genomes. **(B)** In the pie chart, core, soft-core, shell, and cloud genes show inter-strain genomic variation within *A. baumannii*.

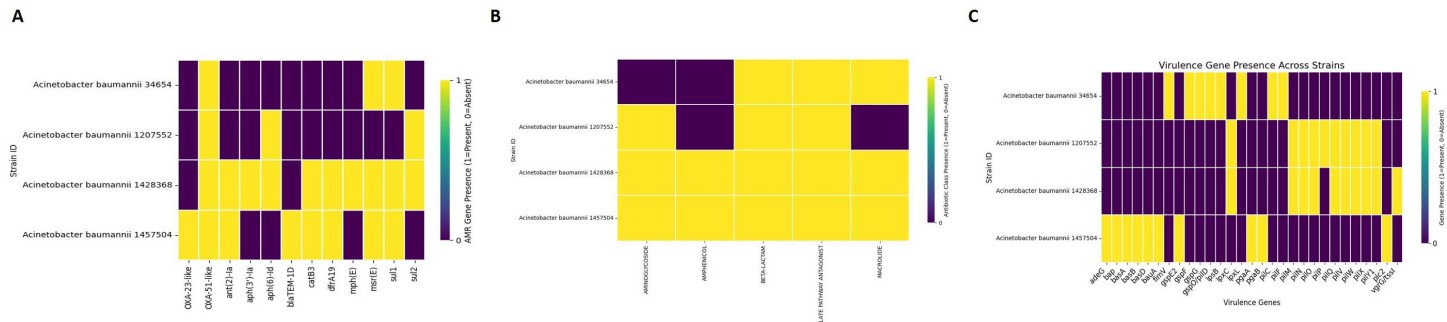

**Fig 3. Visualizing resistance and virulence determinants by heatmap. (A)** Antimicrobial resistance (AMR) related genes. **(B)** Antibiotic class resistance profiles. **(C)** Virulence-associated genes show adaptation mechanisms under antibiotic pressure.

comparison, strain 34654 had fewer AMR genes, inferring a relatively modest resistance potential. AMR genes were then grouped according to their gene classes based on the antibiotic courses discussed in relation to resistance profiles (Fig 3B). These genes confer resistance to β-lactams, macrolides, aminoglycosides, and folate-pathway antagonists. Most significantly, all four strains were found to harbour genes for β-lactam resistance; however, one strain, strain 34654, was found to lack aminoglycoside resistance genes. Such resistance patterns at the class level create significant therapeutic hurdles, especially considering strains with broad resistance profiles. Further virulence gene analysis using the VFDB database indicated strain-specific virulence signatures (Fig 3C). The efflux pump gene *adeG* was conserved in all isolates. In contrast, biofilm-associated genes (*bapA*, *bapB*, *bapC*) were enriched mainly in the perirectal isolates, suggesting a higher colonization potential in gastrointestinal niches. Different components in the Type VI Secretion System or *T6SS*, including *vgrG1* and *hcp*, are shown to exhibit variable and complementary distribution, indicating differential virulence strategies and environmental adaptation between strains (S1 File, Sheet 3).

### 3.5. Principal component analysis of transcriptomic profiles

PCA was employed to elucidate global transcriptional variation among *A. baumannii* strains exposed to sub-MIC levels of four clinically relevant antibiotics (Fig 4): ciprofloxacin (Cip_75), amikacin (Ami_75), meropenem (Mero_75), and polymyxin B (Poly_75). The first two principal components, PC1 and PC2, accounted for 36% and 23% of the total variance, respectively. These numbers distinguish the central axis of transcriptional divergence and stand for the primary sources of variation. Each strain subjected to antibiotic treatment was reported to show distinct clustering for each treatment during the PCA run, indicating that these antibiotics elicit specific transcriptional responses linked to their nature. Samples treated with ciprofloxacin are denoted in red and are most completely depicted by being scattered along both PC1 and PC2 because of their unique and strong transcriptional signature. Amikacin (green) and meropenem (cyan) treatments showed partial overlap, suggesting shared but distinguishable regulatory responses. Samples treated with polymyxin B (purple) formed a discrete cluster, especially along PC2, indicating a very peculiar transcriptional adjustment to this last-resort antibiotic. Minimal intra-treatment variability existed among biological duplicates, emphasizing the highly reproducible and reliable nature of the experimental design. In summary, this study shows that *A. baumannii*, as a pathogen, possesses specific transcriptomic programs developed for various antibiotics, depending on the classes of compounds used and the strain background.

### 3.6. Volcano plot analysis of differential gene expression

Transcriptomic profiling of *A. baumannii* subjected to sub-MIC concentrations of ciprofloxacin, amikacin sulfate, meropenem, and polymyxin B revealed distinct antibiotic-specific transcriptional landscapes, as visualized through volcano

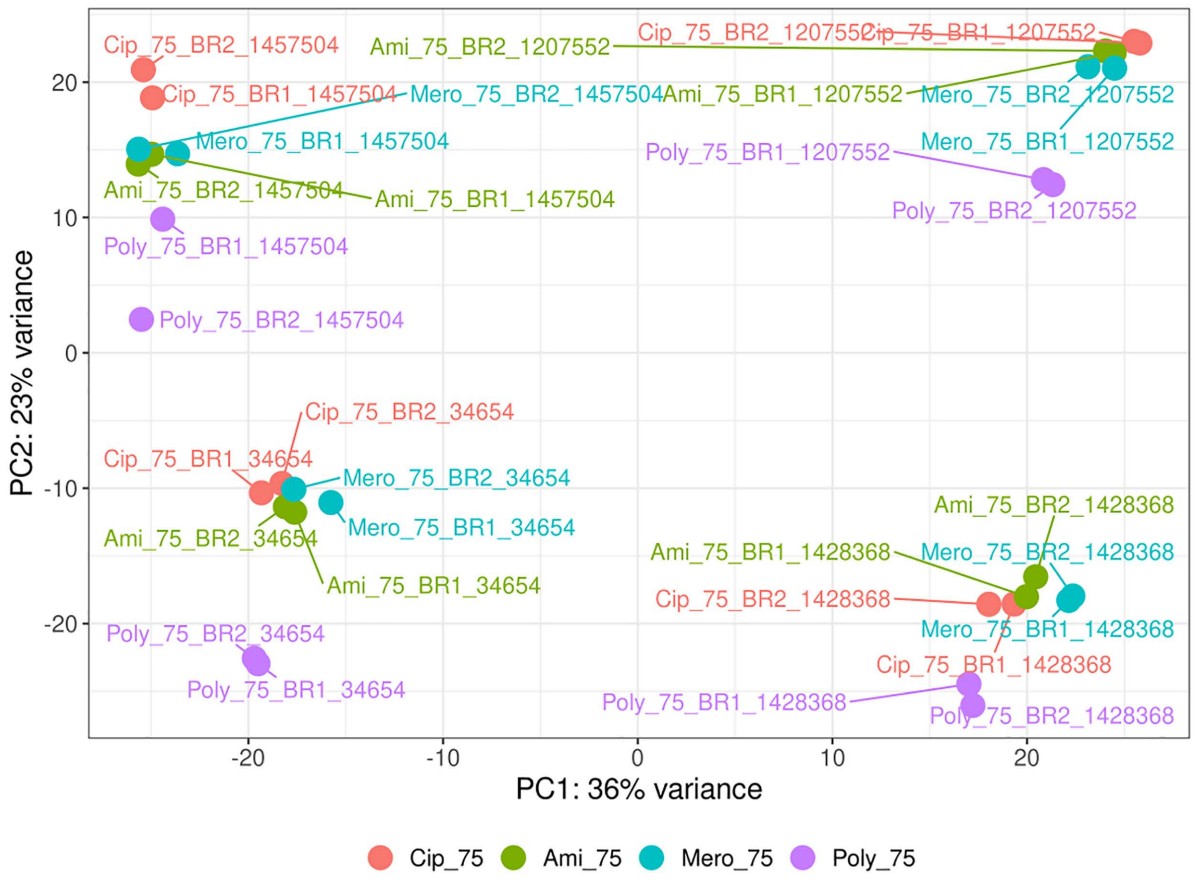

**Fig 4. PCA of the transcriptomic data generated from the four strains of *A. baumannii* (1207552, 1428368, 1457504, 34654) treated with ciprofloxacin, amikacin sulfate, meropenem, and polymyxin B.** Clustering reveals treatment-specific transcriptional signatures.

plots (Fig 5). Differentially expressed genes (DEGs) were identified using stringent statistical thresholds: false discovery rate (FDR) < 0.05 and absolute log$_2$ fold-change (|log$_2$FC|) > 1, ensuring robust detection of biologically relevant transcriptional shifts (Supplementary File 1, Sheets 4–5). Amongst the top 100 significantly downregulated genes, only one gene (0.7%) was found to be consistently downregulated irrespective of the antibiotic treatment, implying that there exists a minimal conserved signature of repression while under antimicrobial stress. The maximum number of genes downregulated uniquely in response to ciprofloxacin treatment was 13 (9.3%); for meropenem, it was 29 (20.7%); and for amikacin sulfate, it was 27 (19.3%). Meanwhile, polymyxin B triggered the lowest number of gene downregulations, with only three genes (2.1%) uniquely downregulated. The results of pairwise and triplet overlap showed significant levels of convergence of the transcriptional repression pathways, suggesting a common but context-dependent regulation. On the other hand, among the top 100 most significantly upregulated genes, 18 transcripts (18.9%) were found to achieve elevated expression across all treatments, representing a conserved adaptive response to stress conditions. An antibiotic-specific transcriptional activation was also exhibited: the highest number of unique upregulated genes was observed with meropenem (11 genes, 11.6%), followed by ciprofloxacin (8 genes, 8.4%), while polymyxin B had less unique activation (1 gene, 1.1%). Shared upregulation signatures mainly included genes implicated in regulating efflux pumps, membrane remodeling, and detoxification of oxidative stress, which are common survival strategies employed by *A. baumannii* under antibiotic pressure. Collectively, these findings delineate a dual-layered transcriptional response comprising both conserved and

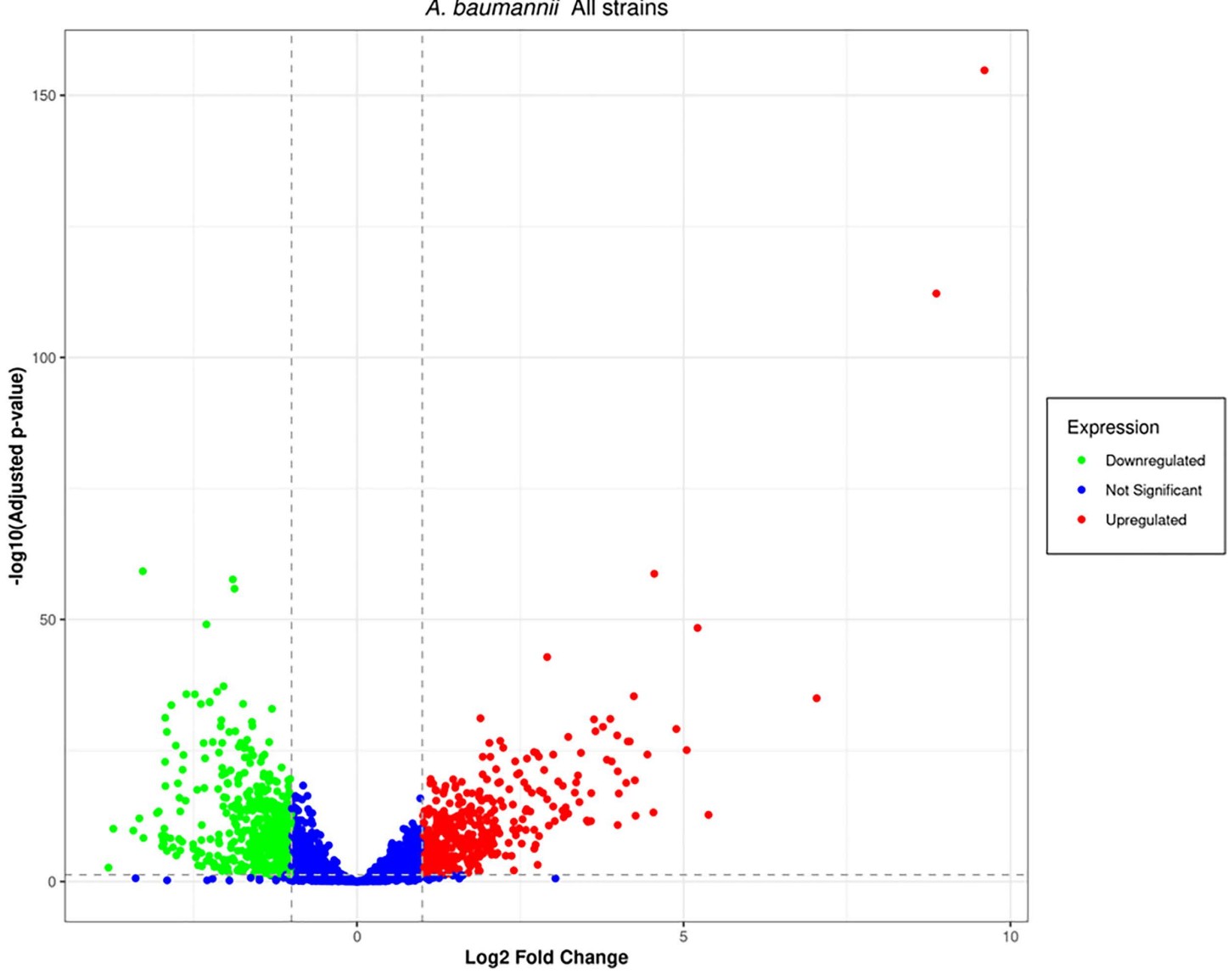

**Fig 5. Volcano plots showing differentially expressed genes post-treatment with antibiotics.** Red: genes significantly upregulated. Green: genes significantly downregulated. Blue: genes below statistical thresholds for log2 fold change or p-value.

drug-specific gene expression programs, with efflux systems and metabolic reprogramming emerging as central themes in antimicrobial adaptation.

### 3.7. Venn diagram-based comparative transcriptomic analysis

To further dissect conserved and strain-specific transcriptional responses, Venn diagram analyses were performed on the top 100 most significantly upregulated and downregulated genes across four *A. baumannii* strains treated with ciprofloxacin, amikacin sulfate, meropenem, and polymyxin B. Fig 6A illustrates the overlap among upregulated genes, in which 18 transcripts representing 18.9% of samples were consistently elevated across all treatment conditions. These conserved genes appear to encode key elements of efflux systems (with emphasis on those from the Mac family), membrane

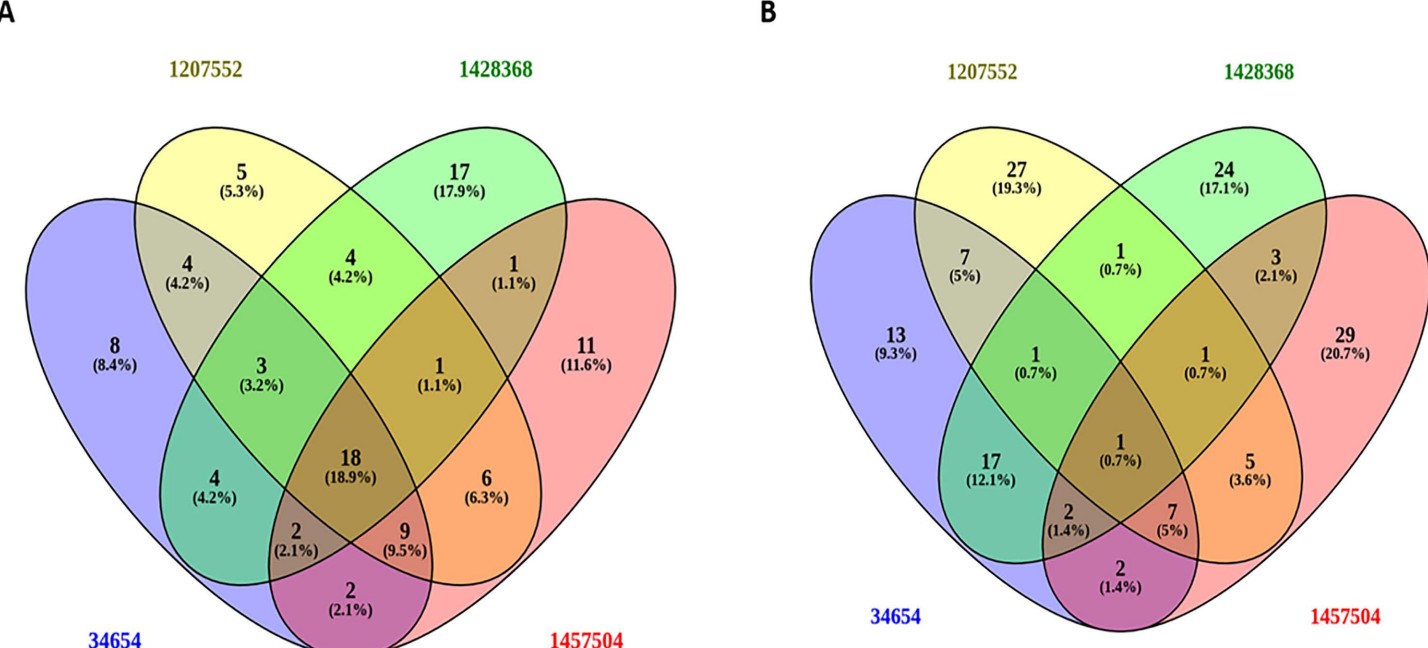

**Fig 6. Venn diagrams showing the overlap of DEGs among the strains.** (A) 100 top upregulated genes. **(B)** Identify 100 top-downregulated genes to reveal conserved transcriptional responses.

integrity modulators, and oxidative stress regulators that would form a core adaptive machinery for survival under different antibiotic challenges. Unique strain-specific transcriptional signatures start appearing, with some overlap between strain pairs and distinct clusters of uniquely activated effectors. This heterogeneity reflects the influence of the genetic background and clinical origin on transcriptional plasticity in *A. baumannii*. Fig 6B presents the Venn diagram for downregulated genes, where only one gene (0.7%) was uniformly suppressed across all strains, highlighting the pronounced variability in transcriptional repression. Downregulated genes predominantly impacted essential cellular processes such as protein biosynthesis, energy metabolism, and DNA replication, suggesting a strategic shift toward metabolic conservation and growth attenuation during sub-MIC antibiotic exposure. All analyses were conducted in triplicate to ensure reproducibility and statistical robustness. The conserved upregulated gene set, particularly those encoding Mac-family efflux pumps, was subjected to virtual screening against the Comprehensive Marine Natural Products Database (CMNPD) to identify potential inhibitory compounds. Promising candidates were subsequently validated through molecular dynamics simulations (MDS), offering mechanistic insights and therapeutic potential for efflux pump-targeted interventions.

### 3.8. Heatmap visualization of conserved differentially expressed genes

Transcriptomic comparison analysis between common DEGs under antibiotic-induced stress was performed to gain insights into the transcriptional landscape of *A. baumannii* in relation to four clinically relevant antibiotics: ciprofloxacin, amikacin sulfate, meropenem, and polymyxin B, all applied in sub-MIC concentrations (Fig 7). The Heat Map of DEGs log fold changes across strains and treatment conditions depicts the magnitude and direction of expression changes. The heat map shows the establishment of two principal clusters of gene expression: an upcluster (red zone) and a downcluster, where gene expression consistently exhibits a positive log2 fold-change. However, this is prominent in the presence of polymyxin B and ciprofloxacin. Canonical resistance-associated elements, such as efflux pump components, molecular chaperones, and oxidative stress response proteins, are found in this cluster, representing classic adaptive resistance

**Fig 7. DEGs tolerant to antibiotic treatment among the four clinical *A. baumannii* isolates.** The color gradient reflects log2 fold change (red = upregulated, blue = downregulated), with hierarchical clustering of co-regulated genes linked to efflux activity, stress adaptation, and resistance phenotypes.

mechanisms. The second cluster (blue zone) embraces downregulated genes with significantly negative-fold changes, especially following stimulation with meropenem and amikacin. These genes are mainly engaged in basic cellular functions, such as central metabolism, ribosome biogenesis, and DNA replication, suggesting that such transcriptional downregulation may be a designed means of energy conservation and persistence against antibiotic stress. The MacA efflux pump subunit, identified in the persistently regulated group, is heavily and evenly upregulated across all strains and treatment conditions, solidifying its role as a major agent of multidrug resistance. In this regard, its status as a priority for clinical intervention in virtual screening and molecular dynamics simulation workflows is justified. Other essential upregulated genes include *RcnB* family proteins, *LolA* (Outer membrane lipoprotein chaperone), *DUF2171 domain-containing proteins, SIMPL domain-containing proteins, and Surface antigen protein 1*. RcnB family proteins, upregulated across conditions, are likely to have roles in metal ion homeostasis and are responsive to oxidative or envelope stress. The outer membrane lipoprotein chaperone, *LolA*, showed mild upregulation in response to polymyxin B treatments, which is consistent with its suspected role in repairing disrupted membranes caused by polymyxin. The regulation of DUF2171 and SIMPL domain-containing proteins differs among the tested strains, indicating that the strains display varying adaptive mechanisms. Surface antigen protein one is also upregulated in response to several different treatments, possibly functioning to support immune evasion or membrane remodeling in the context of the resistance phenotype. In condition-specific responses, polymyxin B induced the most significant upregulation of stress-related and membrane-associated genes, a pattern consistent with its mechanism of action of outer membrane damage. The downstream effects of meropenem and amikacin treatments showed a wide downregulation of genes contributing to biosynthesis and cell division, likely referring to a metabolic slowdown as a potential persistence strategy. Thus, the transcriptomic landscape indicates that a resistance mechanism in *A. baumannii* is conserved but modulated dynamically, with the efflux systems, in particular MacA, being the prime actors. The heatmap allows a high-resolution view of the dynamics related to gene-expression remodeling in *A. baumannii* under different antibiotic pressures, information that can be translated into therapeutic applications.

## 3.9. Gene ontology (GO) enrichment analysis of upregulated genes

To explore the functional roles of the top 100 commonly upregulated genes across all four antibiotic treatments, we performed GO enrichment analysis spanning three categories: Biological Process (BP), Molecular Function (MF), and Cellular Component (CC) (Fig 8). This allowed for a more complete understanding of transcriptional adaptations that underlie the *A. baumannii* response to sub-MIC antibiotic stress.

### 3.9.1. Biological processes (BP).
The analysis has demonstrated a significant enrichment of metabolic and stress-response pathways. Purine metabolism, including adenosine or inosine turnover, along with salvage or catabolic activities, represents a level of enrichment suggesting that stress conditions are associated with increased nucleotide recycling *in vivo*. Upregulation of amino acid metabolic pathways, particularly those involving gamma-aminobutyric acid (GABA), aspartate, leucine, and glutamine, indicates a strategically reconfigured metabolism in the cell to drive survival. Several enriched genes involved in xenobiotic detoxification, oxidative stress response, and DNA repair further indicate that defense mechanisms are activated against antibiotic exposure. Other upregulated pathways included lipid and fatty acid metabolism, protein processing, and signaling relating to antibiotic resistance.

### 3.9.2. Molecular functions (MF).
The molecular functions that were highly enriched were basically derived from their enzymatic activities, such as decarboxylases, dehydrogenases, oxidoreductases, and transferases. Thus, the EMS-generated functions showed an increase in metabolic flux and redox activity during antibiotic challenge. The binding functions from various proteins were also upregulated, particularly for binding to metal ions, ATP, and cofactors, indicating increased energy needs and regulatory enzyme activation. Upregulated functions related to transport were mainly efflux and membrane translocation, reflecting is known about the bacterium's resistance strategies.

### 3.9.3. Cellular components (CC).
There was a great representation among the GO terms with respect to membrane structures and transport complexes. These were transmembrane transporter complexes, components of efflux pumps, and

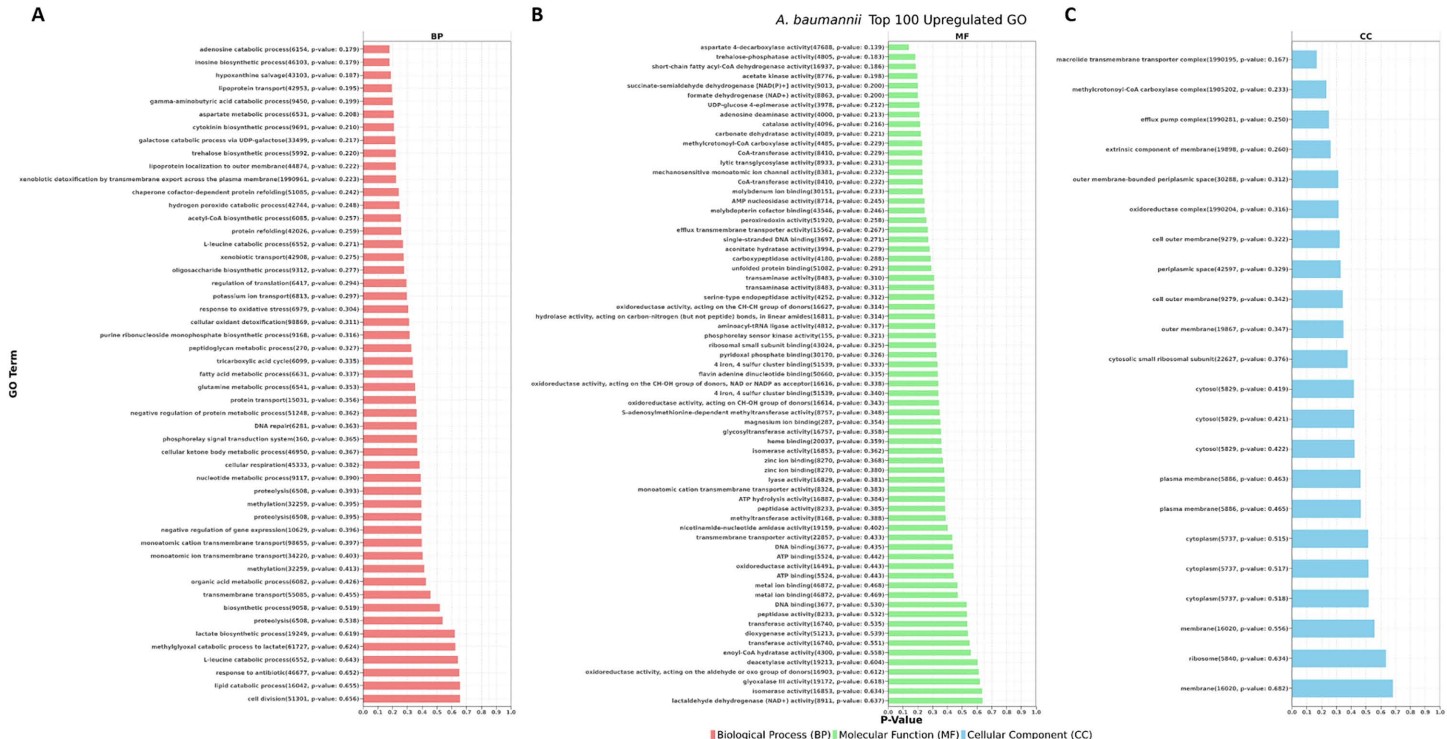

**Fig 8. Gene Ontology (GO) enrichment based on the top 100 commonly upregulated DEGs. (A)** Biological Processes **(B)**, Molecular Functions **(C)**, Cellular Components: Enriched terms point to pathways linked to the adaptation to antibiotic stress.

especially the outer membrane, thus emphasizing how essential membrane dynamics are for bacterial adaptation. Enrichment also included periplasmic space and cytosolic components in addition to ribosomal subunits, pointing to active protein synthesis and internal restructuring. These results matched a strong cellular response aimed at homeostasis and combating antibiotic action. Collectively, this evidence indicates that *A. baumannii* adapts by changing its metabolism, activating efflux systems, and improving cellular stress. The increase in membrane transport and detoxification pathways provides proof that the Mac family of efflux pumps drives these adaptive responses and represents promising therapeutic targets.

### 3.10. Structural validation via ramachandran analysis

The conformational quality of the MacB homology model has been further ascertained using Ramachandran plot statistics, which concerned 1,164 residues, excluding glycine and proline due to their exceptional flexibility with respect to the backbone. The analysis revealed that 92.8% (1,080 residues) reside within the most favored regions [A, B, L], while 5.7% (66 residues) occupy the additionally allowed regions [a, b, l, p]. A minor fraction, 1.2% (14 residues), falls within the generously allowed zones [~a, ~b, ~l, ~p], and only 0.3% (4 residues) are located in disallowed regions [XX] (Fig 9). These metrics confirm the stereochemical robustness of the model, with over 90% of residues adopting energetically favorable conformations. This high-quality structural profile substantiates the model's suitability for downstream computational analyses, including virtual screening and rational drug design.

### 3.11. High-throughput structure-based virtual screening

Structure-based virtual screening (SBVS) is a reliable and crucial method in computational drug development that uses protein-ligand interaction simulations to find bioactive molecules. The methodology is generally known as molecular

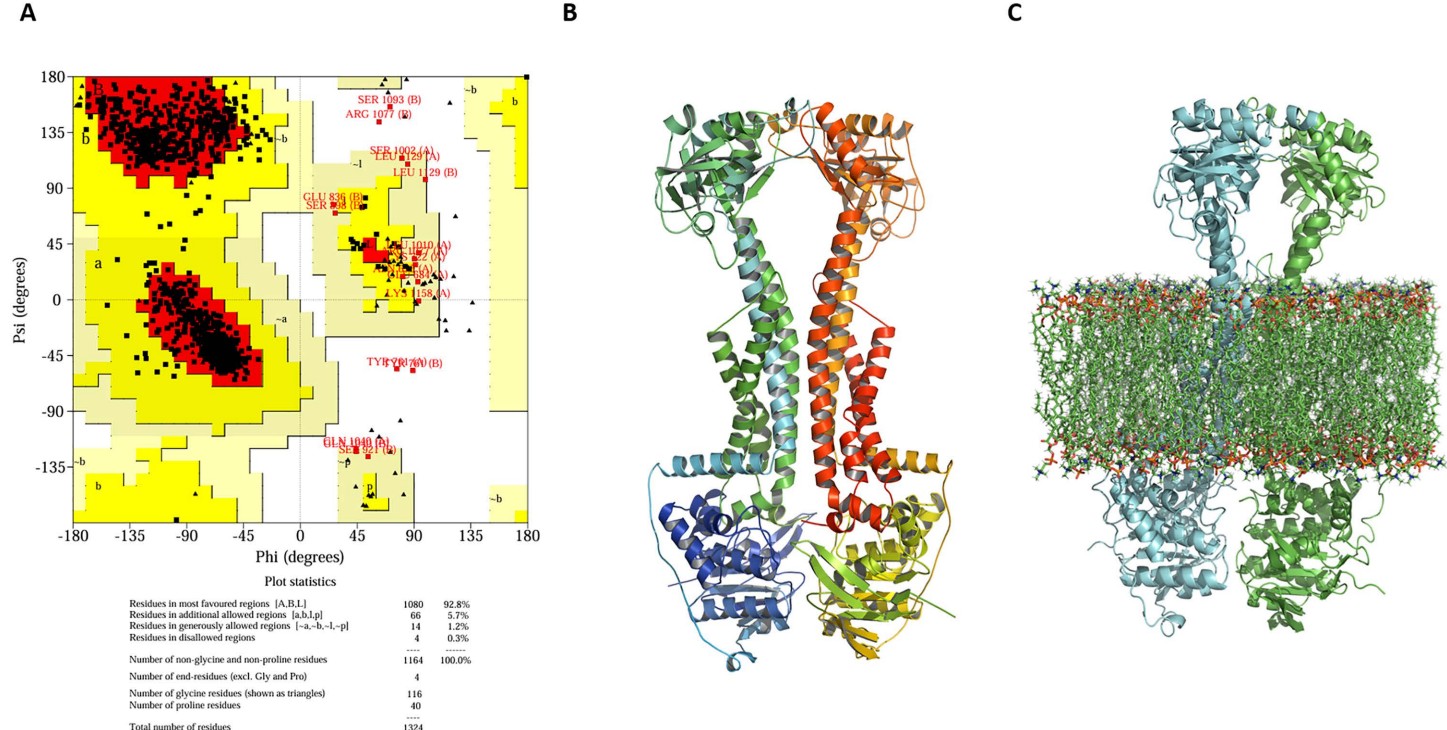

**Fig 9. MacB efflux pump structural modeling and validation. (A)** Ramachandran plot confirming stereochemical integrity. **(B)** The tertiary structure predicted for MacB. **(C)** Integration of MacB into a POPC lipid bilayer model for membrane-contextualized simulations.

docking, a method for predicting the optimal position and affinity of small molecules in the active site of the target protein. These procedures can be used for prioritizing ligands based on the predicted binding energies or inhibition constants. Herein, high-throughput virtual screening has been conducted against the entire structural surface of MacB and restricted to defined binding pockets. A docking study utilized a curated library of 31,561 marine-derived natural products from the CMNPD database against MacB to find possible inhibitors. Among these, CMNPD27284 emerged as a top candidate as it had the most favorable binding energy of −7.20 kcal/mol with four hydrogen bonds to critical residues GLN759, ASP835, and ARG760. Comparative docking of this compound with reference antibiotics gave the following results: Ciprofloxacin: Binding energy of −6.23 kcal/mol; one hydrogen bond with ARG1327. Amikacin sulfate: Binding energy of +1.83 kcal/mol; four hydrogen bonds with HIS762, ARG760, and HIS868. Meropenem: Binding energy of −4.93 kcal/mol; two hydrogen bonds with ALA840 and ARG760. Polymyxin B: Highly unfavorable binding energy of +70.83 kcal/mol; no hydrogen bonding interactions observed. Compared with all control drugs, CMNPD27284 showed the maximum binding affinity, with ciprofloxacin coming second among the tested compounds. These findings warrant an effective experimental quest to further prove the workability of CMNPD27284 as a potential lead compound. Interaction profiles are shown in both 2D and 3D in Fig 10.

### 3.12. ADMET profiling of CMNPD27284

Three-dimensional pharmacodynamics and physicochemical description of the marine-derived CMNPD27284 with a molecular weight of 276.24 g/mol are summarized in Table 2. The molecule with three hydrogen bond donors and six hydrogen bond acceptors gives a total polar surface area (TPSA) of 104.06 Å$^2$, which is a parameter indicative of solubility and permeability characteristics favorable for drug development. Following the docking phase, CMNPD27284 was

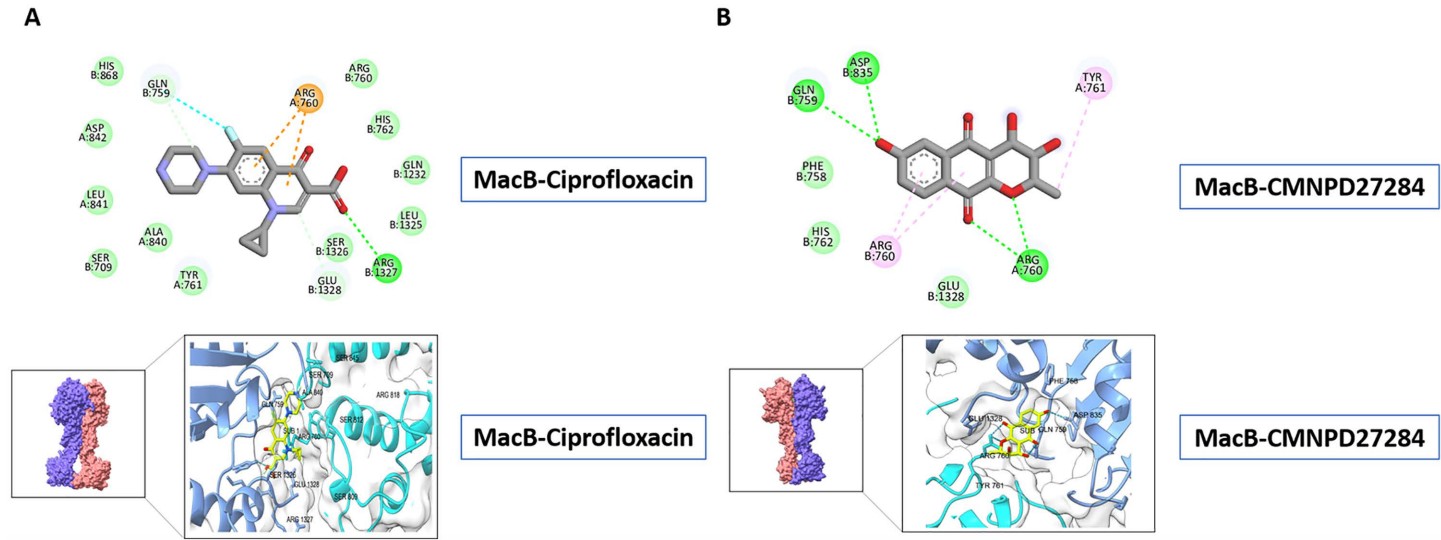

**Fig 10. MacB interaction with CMNPD27284 and ciprofloxacin in molecular docking.** 2D/3D diagrams emphasize key binding residues, interaction energies, and ligand orientation within the efflux channel.

**Table 2. The physicochemical characterization of CMNPD27284, along with docking-derived measures against the MacB efflux pump, serves as evidence supporting its potential as a lead compound.**

| Phytocompound | Molecular weight (g/mol) | H-bond donor | H-Bond acceptor | Tpsa (A²) | Rotatable bonds |
|---|---|---|---|---|---|
| CMNPD27284 | 276.24 | 3 | 6 | 104.06 | 0 |

subjected to molecular dynamics simulations to further analyze its stability and interaction dynamics under physiologically relevant conditions.

## 3.13. Molecular dynamics simulations (MDS)

MDS were conducted to elucidate the atomistic behavior of the MacB protein and its ligand-bound complexes over an extended temporal scale. This approach studied the dynamic features of the protein-ligand assemblies, including structural integrity, stability of interactions, and solvated adaptational conformation. Structural descriptors measured in this study were root mean square deviation (RMSD), root mean square fluctuation (RMSF), radius of gyration (Rg), solvent-accessible surface area (SASA), and intermolecular hydrogen bonding (Supplementary File 1, Sheets 6–9). To obtain a holistic interpretation of size, folding, and interaction robustness, MDS were conducted for up to 1000 nanoseconds on the apo form of MacB, its complex with the marine-derived compound CMNPD27284, and with the reference antibiotic ciprofloxacin.

### 3.13.1. Root mean square deviation (RMSD).
RMSD serves as a quantitative metric for assessing the conformational stability of macromolecular systems (Fig 11A). The average RMSD of MacB in a dynamic environment, when compared with other transmembrane transporters, is $0.72 \pm 0.15\,nm$. This indicates a certain flexibility in the structure. The RMSD of MacB-ciprofloxacin, which is a ligand-binding complex, is reduced to $0.55 \pm 0.13\,nm$, indicating that the conformer made a way to undergo rigidity and stabilization while acting strongly to allow ligand-residue interactions. On the other hand, the MacB–CMNPD27284 complex showed an RMSD of $0.69 \pm 0.13\,nm$, indicating a somewhat more flexible conformation. This elevated RMSD implies that CMNPD27284 permits localized structural rearrangements, potentially facilitating deeper accommodation within the efflux cavity.

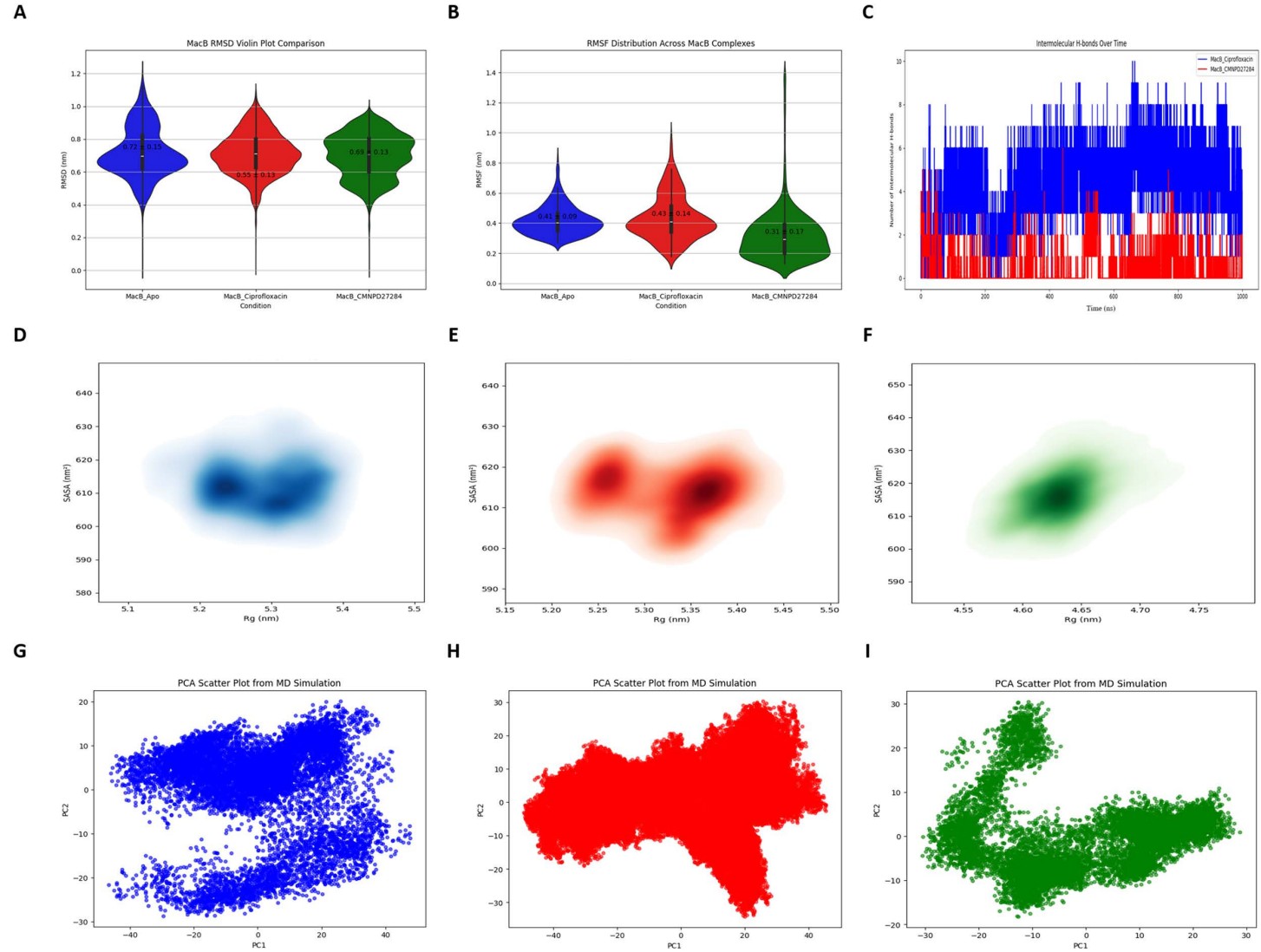

**Fig 11. MacB-ligand complexes experimental molecular dynamics measures. (A)** Violin plots of RMSD distributions for apo (blue), MacB-ciprofloxacin (red), and MacB-CMNPD27284 (green) states. **(B)** RMSF profiles among residues. **(C)** The number of hydrogen bonds was counted over the overall simulation time. **(D-F)** KDE plots of conformational sampling in apo, MacB-ciprofloxacin, and MacB-CMNPD27284 bound states. (G-I) PCA scatter plot depicting the prevailing movements of the three binding conditions.

### 3.13.2. Root mean square fluctuation (RMSF).

RMSF serves as a critical metric for quantifying the residue-level dynamic behavior of protein side chains throughout molecular dynamics simulations (Fig 11B). RMSF provides insight into local conformational mobility in relation to flexible areas in loops and solvent-exposed transmembrane domains. The average RMSF across surface-accessible regions of the averaged apo form of the MacB transporter was $0.41 \pm 0.09$ nm, indicating a definite measure of flexibility in those regions. The finding of this study suggests that the RMSF for the MacB-ciprofloxacin complex was nearly $0.43 \pm 0.14$ nm, showing a slightly elevated RMSF upon ligand-binding. Some domains, therefore, are stable while others allow movement. However, there is a potential binding site in MacB-CMNPD27284, whose associated RMSF value is low ($0.31 \pm 0.17$ nm), indicating a greater degree of local rigidity. The CMNPD27284 has exerted conformational restraint on critical residues both within and outside the binding cavity, significantly suppressing

atomic fluctuations. This kind of stabilization could facilitate the favorable interaction of the ligand with the functional domains of MacB, thus avoiding the necessary structural rearrangements for efflux activity. The RMSF profile thus supports the hypothesis that CMNPD27284 confers targeted rigidity across critical transporter regions, underscoring its specificity and potential as a robust efflux pump inhibitor.

### 3.13.3. Hydrogen bond (H-Bond) analysis.

H-bonds are crucial for all structural coherence and thermodynamic stability of protein-ligand complexes during simulations. Thermodynamically, these non-covalent interactions contribute to conformational stability and help lock the ligand into binding at the interface. During 1000 ns of simulation time, the ligand-bound systems, viz. MacB-ciprofloxacin and MacB-CMNPD27284 exhibited stable H-bond networks that fluctuated slightly, possibly indicating adaptive repositioning in a dynamic binding pocket (Fig 11C). In fact, the system associated with MacB-CMNPD27284 maintained six hydrogen bonds under the MacB-ciprofloxacin complex, in which the average number was ten hydrogen bonds forming and breaking. Although it had fewer hydrogen-bonding interactions with MacB, it occupies beneficial spatial sites in the transmembrane and periplasmic regions of MacB that are pivotal for the regulation of efflux. These anchor positions compensate to some extent for lower-numbered bonds by increasing hydrophobic interaction, further stabilizing the complex. The persistence and functional relevance of hydrogen bonds suggest that CMNPD27284 has both energetically viable and structurally relevant interactions with MacB. This is likely to prevent the conformational transitions required for substrate extrusion, making it a perfect candidate as a selective and potent efflux pump inhibitor.

### 3.13.4. Radius of gyration (Rg) and solvent accessible surface area (SASA).

The Rg quantifies the spatial compactness of a protein, serving as a proxy for structural integrity during simulation. It was calculated as the mass-weighted RMSD of atomic positions relative to the center of mass. The apo MacB protein exhibited an average Rg of $5.28 \pm 0.06$ nm, reflective of a stable, yet conformationally permissive architecture. A marginal increase to $5.32 \pm 0.05$ nm was observed for the MacB-ciprofloxacin complex, whereas for the MacB-CMNPD27284 bound complex, a significant decrease was observed at $4.63 \pm 0.03$ nm. Such contraction indicates a ligand-induced compaction that may restrain internal flexibility and stabilize the transporter against conformational changes required for efflux. SASA is a measure of solvent exposure of amino acid residues and, conversely, an indication of conformational openness. The apo MacB form exhibited an average SASA of $612.63 \pm 8.42$ nm²; the MacB-ciprofloxacin complex exhibited an extra slight elevation at $613.93 \pm 6.74$ nm². The MacB-CMNPD27284 complex was blown up further to $616.57 \pm 8.61$ nm², probably indicating localized rearrangements at the surface to accommodate the ligand. However, the increased area does not seem to affect the overall compactness, as indicated by the lowered Rg values. Altogether, analyses of Rg and SASA show that MacB-CMNPD27284 enforces a compact structure that remains dynamic, thus supporting its functional relevance in efflux inhibition. Kernel density estimation (KDE) plots for SASA-surface area versus Rg for all three systems are shown in Fig 11D-11F.

## 3.14. Principal component analysis (PCA)

PCA was employed to dissect the collective atomic motions associated with ligand binding in MacB systems (Fig 11. G–I). By diagonalizing the covariance matrix of the atomic positional fluctuations, PCA provides eigenvectors and corresponding eigenvalues, which elucidate the dominant motion trajectories. Significant changes usually occur at the shifted positions of two principal components, where most of the transitions are performed during the shifts of substrate translocation and in the accommodation of the ligand. The absence of any ligands in the apo states of MacB revealed broad sampling conformations, demonstrating its intrinsic flexibility and dynamic heterogeneity under the unconstrained conditions created by a ligand. With MacB-ciprofloxacin binding, the conformational landscape was reduced, indicating that much of the flexibility was masked, thus conferring some stabilization upon the system. The most confined conformational space characterized the complex of MacB-CMNPD27284, suggesting that this marine-derived ligand imposes the strongest conformational restraints. That confinement may well reflect a ligand-induced stabilization mechanism locking the transporter

in a compact conformational ensemble and blocking the large-amplitude motions needed for efflux function. PCA findings converge well with RMSD, RMSF, and Rg, altogether reinforcing the view that CMNPD27284 confers rigidity and stabilizes MacB.

## 3.15. Free energy landscape (FEL)

FEL analysis was performed for all three systems, apo MacB, MacB-ciprofloxacin, and MacB-CMNPD27284, with mapping of the conformational energy states along the first two principal components (PC1 and PC2) (Fig 12). From the thermodynamic perspective of protein dynamics, FEL indicates the energy minima and transition states of conformational stability and perturbations caused by ligand binding. Apo MacB presented multiple shallow basins of energy and more frequent transitions, both suggesting a high conformational entropy and dynamic flexibility. On the contrary, the MacB-ciprofloxacin complex showed comparatively fewer transition states and moderately deep energy minima, which suggests partial stabilization upon ligand binding. In a most interesting development, the MacB-CMNPD27284 complex displayed a single dominating energy basin with a steep and pronounced global minimum, indicating a very stable conformational state with slight fluctuations in energy. The deep and narrow appearance of this energy basin suggests that the ligand binding has strongly stabilized the MacB, effectively locking it in a low-energy conformation and preventing any of the dynamic rearrangements that may be required for efflux activity. These FEL results further strengthen the structural information derived from RMSD, Rg, and PCA and consequently support that MacB-CMNPD27284 is a potential efflux pump inhibitor.

## 3.16. Dynamic cross correlation matrix (DCCM)

The DCCM analysis was performed to elucidate the correlated and anticorrelated motions among MacB residues in both ligand-bound and unbound states (Fig 13). Positive correlations, in red, indicate synchronized movements, while negative correlations, in blue, indicate movements in opposing directions that are commonly associated with allosteric signaling and coordination of domains. Correlated motion in apo MacB, particularly between 320–520 and 980–1190, is relevant to the flexible loops and periplasmic parts of the molecule, which are involved in possible substrate transport. The binding of the ligands to both MacB-ciprofloxacin and MacB-CMNPD27284 showed a substantial decrease in correlated motion, indicating that ligand binding has a damping effect on the internal flexibility of MacB, thus hampering the dynamic coupling between the domains. Correlated and anticorrelated motions of MacB-CMNPD27284 were more significant than those for ciprofloxacin, indicating that the binding would induce higher structural coherence and rigidification. Such close-fitting

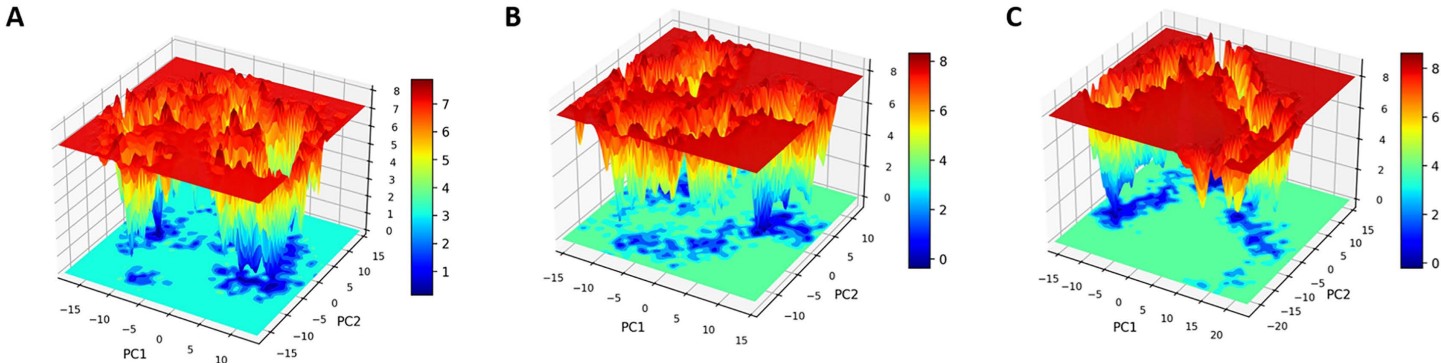

**Fig 12. Free-energy landscape (FEL) analysis of MacB conformers. (A)** Apo form **(B)** MacB-ciprofloxacin **(C)** MacB-CMNPD27284 Energy minima and basin topology reveal ligand-induced stabilization.

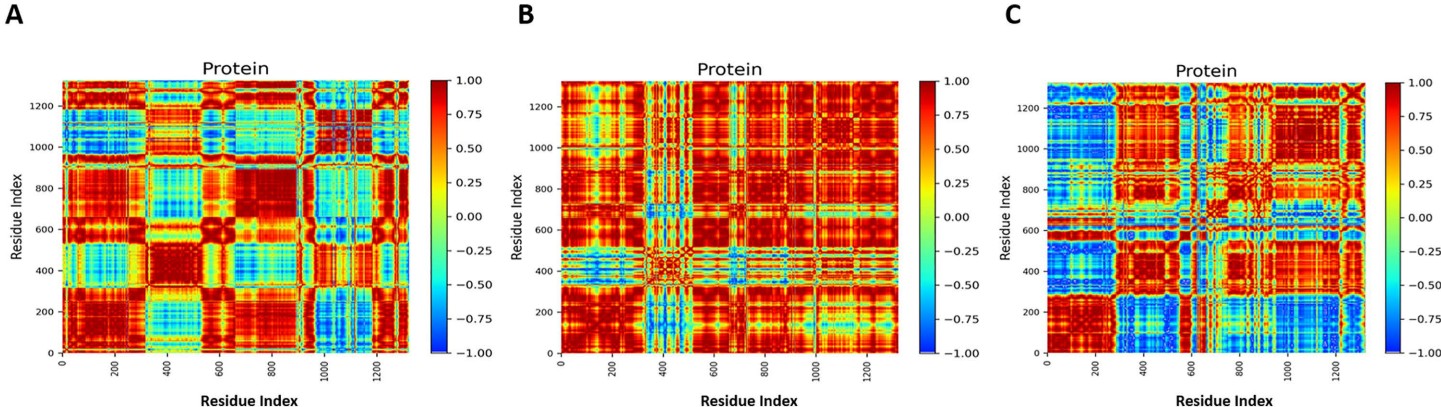

**Fig 13. Dynamic cross-correlation matrix (DCCM) analysis of residues' motions in MacB complexes.** (A) Apo (B) MacB-ciprofloxacin (C) MacB-CMNPD27284 correlation maps reveal ligand-dependent modulation of structural dynamics and flexibility.

coupling of residues implies a more compact conformation, dynamically stabilized by MacB-CNMPD27284, which is detrimental to the functional mobility required for efflux. That DCCM data support the contention that MacB-CMNPD27284 binding would restrict interdomain mobility, thereby affording conformational stability, although at a cost of decreasing transporter activity.

### 3.17. MM-PBSA analysis

To quantify the thermodynamic favorability of ligand binding, MM-PBSA calculations were performed across the simulation trajectory (Table 3). This method combines molecular mechanics with solvation models, taking into account Van der Waals interactions, electrostatics, as well as polar and non-polar solvation contributions. The binding free energy of the MacB-ciprofloxacin complex was found to be −28.834 ± 44.331 kcal/mol, whereas MacB–CMNPD27284 displayed a more favorable energy of −30.813 ± 15.326 kcal/mol. Thus, the more negative binding energy value found for MacB-CMNPD27284 suggests a stronger and more stable interaction with MacB compared to the reference drug. The lower standard deviation of the data for MacB-CMNPD27284 indicates that the compound displayed more consistent binding to MacB throughout the simulation. The energetics most likely arise from the concerted effect of Van der Waals and electrostatic forces, which stabilize the ligand inside the binding cavity. Overall, MM-PBSA results further confirm the molecular dynamics data in favor of MacB-CMNPD27284, which exhibits high binding affinity and thermodynamic stability, substantiating it as a perfect candidate for a MacB-targeted efflux pump inhibitor.

### 4. Discussion

The present investigation explores the transcriptomic landscape of *Acinetobacter baumannii* ST2 (International Clone II) when exposed to sub-lethal concentrations of four clinically relevant antibiotics: ciprofloxacin (first-line), amikacin sulfate

**Table 3. MM-PBSA-derived binding free energy estimations for MacB-CMNPD27284 and MacB-ciprofloxacin within the MacB complex, benefiting our understanding of thermodynamic stability and ligand efficiency.**

| Targeted protein with ligand complex | Van der Waal Energy (kJ/mol) | Electrostatic Energy (kJ/mol) | Polar solvation Energy (kJ/mol) | SASA Energy (kJ/mol) | Total Binding Energy (kJ/mol) |
|---|---|---|---|---|---|
| MacB with Ciprofloxacin | −89.192 +/- 9.038 | −525.193 +/-52.012 | 602.029 +/-42.702 | −16.477 +/-1.020 | −28.834 +/-44.331 |
| MacB with CMNPD27284 | −82.282 +/-14.206 | −71.628 +/- 24.495 | 135.203 +/-30.923 | −12.106 +/-1.329 | −30.813 +/-15.326 |

(second-line), meropenem (third-line), and polymyxin B (last-resort). The study underscores the dynamic regulation of antibiotic-induced stress responses, with a particular emphasis on efflux pump-mediated resistance mechanisms [57]. A pangenomic analysis showed that *A. baumannii* has a relatively conserved genomic core (59.2%), in accordance with the fact that it lacks soft-core and cloud genes, and serves as a genetically stable backbone (Fig 2). Genetically close to each other, strains clustered into separate groups based on the isolation source perirectal versus sputum suggesting host-specific evolutionary pressures. The most salient findings highlight essential virulence factors and resistance-associated genes, such as the efflux-related gene *adeG* [58]. Basically, the Mac family of efflux pump systems is crucial in bacterial stress responses to antibiotics and thus plays a critical role in conferring further resistance [59].

Apart from MacB, other resistance mechanisms also play an essential role in establishing multiple resistances in *A. baumannii*. For instance, the efflux systems *AdeABC*, *AdeIJK*, and *AdeFGH* belong to the RND-type efflux pumps [60]. Together with MacB, these efflux pumps contribute to the increased efflux of various antibiotics from the cell and thus limit their intracellular accumulation [61]. Simultaneously, enzymatic degradation of β-lactams is facilitated by β-lactamases, such as the OXA-type carbapenemases and NDM metallo-β-lactamases, which inactivate these drugs [62]. Furthermore, various factors damage or alter outer membrane porins, reducing drug permeability and limiting antibiotic entry [63]. All these mechanisms combine to form a complex network of resistance. Efflux, enzyme activity, and membrane changes enable bacteria to survive when they encounter antibiotics [64]. Besides this, biofilm-associated genes and genes associated with type VI secretion systems (T6SS) could further contribute to strain-specific pathogenesis that culminates in enhanced virulence of the bacteria [65]. Multivariate computational analysis techniques, PCA, volcano plots, heat maps, and Venn diagrams were employed to study the adaptation of *A. baumannii* transcriptomics under induced stress by antibiotics. Using PCA, we observed significant changes in gene expressions, resulting in a unique transcriptional signature for each antibiotic (Fig 5). Polymyxin B has a highly different response compared with that of ciprofloxacin. Amikacin sulfate and meropenem treatments characterized a reciprocal but partial transcriptional overlap in effect, signifying shared stress responses with slight strain-based differences. High volcano mapping revealed differentially expressed genes (DEGs), leading to a core of 18 genes upregulated in all antibiotic treatment conditions, primarily related to efflux pump activities and stress response (Fig 6). Of great interest, the one such consistently downregulated gene across strains indicated strain-specific repression. The deductions above suggest that *A. baumannii* has devised several conserved defenses that together augment its potency against antibiotics by activating efflux pumps, oxidative stress responses, and membrane remodelling [17]. The heatmap analysis of the transcripts clusters them into two sets: upregulated stress resilience genes (associated with efflux pumps, chaperones, and stress-related proteins) and downregulated genes associated with essential pathways of metabolism and DNA replication machinery [66]. Nevertheless, among the mechanisms of resistance found in the various treatments with antibiotics, the MacA efflux pump subunit is an essential candidate for a novel therapeutic intervention. Its expression significantly increased a variety of membrane-associated genes in response to polymyxin B [67].

Conversely, treatment with meropenem and amikacin sulfate resulted in the suppression of biosynthetic and cell division-associated genes, thus indicating metabolic slowing as a persistence strategy. GO analysis of the top hundred upregulated genes revealed significant biological processes, such as purine and amino acid metabolism, and oxidative stress responses (Fig 8), which are manifestations of metabolic reprogramming as an adaptive mechanism [68]. Most of the molecular functions involving enzymatic activity were upregulated in the efflux pumps, whereas a few exceptions involved redox activities and transport systems [69]. Therefore, cellular component enrichment analysis as above indicates the importance of dynamic membranes, efflux complexes, and ribosomal subunits towards adaptive responses under antibiotic stress. [70]. Accepting that efflux plays a central role in resistance, the study also extended the scope into high-throughput virtual screening (HTVS), this time utilizing the CMNPD database for identification of promising marine-derived compounds that potentially inhibit the MacB efflux pump. Among all others, the most outstanding candidate MacB-CMNPD27284 was found to be uniquely endowed with a much better binding affinity to MacB protein than

conventional antibiotics such as ciprofloxin, amikacin sulfate, meropenem, and polymyxin B. Binding energy calculations revealed that the marine compound had a binding energy of −7.20 kcal/mol, forming four hydrogen bonds with critical residues (GLN759, ASP835, and ARG760), thus indicating the binding would be adequate for consideration as drug candidate (Table 4).

Enhanced testing for stability and possible interactions of MacB-CMNPD27284 were carried out using MDS (Fig 11). Therefore, the simulations provided valuable information regarding the conformational flexibility and stability of the MacB-ligand complex bindings [71]. As indicated by RMSD analysis, binding of the MacB-CMNPD27284 complex led to a very slight increase in the flexibility of the protein [72] in comparison to the free MacB protein, suggesting possible changes in structure upon binding of the ligand. The RMSF analysis [73] suggested regions of MacB associated with pump activity became more flexible with the addition of compound CMNPD27284. Also, Rg studies revealed a minor compaction [74] of the MacB framework after ligand attachment, possibly indicating that the folded conformation is more stable. The SASA analysis showed very slight increases in the solvent-exposed surface areas of both MacB-ciprofloxacin and MacB-CMNPD27284 complexes, which probably reflects some expansion of the protein structure [75]. H-bond analysis indicated that while seven hydrogen bonds were formed in the MacB-ciprofloxacin complex, the MacB-CMNPD27284 complex formed only five hydrogen bonds, suggesting that these two complexes are relatively strong but less stable. As observed from the PCA analysis of the MacB-ligand complexes, it can be inferred that despite the apo conformation of MacB occupying a huge conformational space, the MacB-CMNPD27284 complex possessed a slightly larger conformational state when compared to the ciprofloxacin complex. This suggests that CMNPD27284 can elicit a different conformational change in MacB, as that alteration can affect the activity of this efflux pump. The analysis of the FEL indicated that the apo form of MacB has multiple transition states that end with well-defined global energy minima, thus ensuring very high stability [76]. On the contrary, the MacB-ciprofloxacin complex exhibits multiple transition states with moderate global energy minima, indicating lower stability.

Nevertheless, the MacB-CMNPD27284 complex displayed only a single, well-defined global energy minimum, suggesting that it was the most stable among the complexes. This indicates that CMNPD27284 is a more stable inhibitor of the efflux pump MacB. The DCCM analysis ultimately indicated that ligand binding lowered the correlation motions [77] of MacB residues in the 320−520 and 980−1190 regions, as they displayed negative correlations, thus indicating more stability in ligand-bound complexes. The MMPBSA study of binding free energies confirmed that the affinity [78] of MacB-CMNPD27284 (−30.813±15.326 kcal/mol) was higher than that of MacB-ciprofloxacin (−28.834±44.331 kcal/mol), which makes it a better therapeutic candidate against the MacB efflux pump.

## 5. Conclusion

This study highlights the flexibility of *Acinetobacter baumannii* to antibiotic-induced stress, with efflux pump activity being the central mechanism of resistance. All-encompassing genome-wide transcriptome studies may dissect networks of genes involved in efflux transporters, combating oxidative stress, and membrane restoration. These are, to varying

**Table 4. Comparison of binding affinities and H-bonded interactions between conventional antibiotics (namely, ciprofloxacin, amikacin sulfate, meropenem, and polymyxin B) and marine-derived medicaments, based on triplicate docking results.**

| Protein with Ligand complexes | Dock 1 (kcal/mol) | Dock 2 (kcal/mol) | Dock 3 (kcal/mol) | Average (kcal/mol) | No. of H-Bond | H-Bond Interactive Residue |
|---|---|---|---|---|---|---|
| MacB-Ciprofloxacin | −6.3 | −6.2 | −6.2 | −6.23 | 1 | ARG1327 |
| MacB-Amikacin sulfate | 4.1 | −0.1 | 1.5 | 1.83 | 4 | HIS762, ARG760, HIS868 |
| MacB-Meropenem | −5 | −4.9 | −4.9 | −4.93 | 2 | ALA840, ARG760 |
| MacB-Polymyxin B | 72.2 | 67.1 | 73.2 | 70.83333 | Nil | Nil |
| MacB-CMNPD27284 | −7.2 | −7.2 | −7.2 | −7.20 | 4 | GLN759, ASP835, ARG760 |

extents, hallmarks of activation by the bacterium's complex arsenal of defenses. Indeed, marine-derived CMNPD27284 was later found to be a potent MacB inhibitor through the combination of high-resolution RNA-Seq data with high-throughput virtual screening (HTVS) and molecular dynamics simulations (MDS). Binding and conformational stability are better for MacB-CMNPD27284 as compared to conventional antibiotics. Such limitations in the exclusive use of publicly available genomic datasets arise from their inability to capture heterogeneity or resistance phenotypes that might exist within clinical *A. baumannii* isolates, which would impede their translation into clinical use. The same definitely applies to an absence of experimental validation regarding the *in vivo* efficacy and pharmacodynamics of CMNPD27284. Bridging these gaps warrants future studies to include phenotypic assays with carbapenem-resistant clinical isolates as well as *in vivo* infection models, such as murine systems, to consolidate the therapeutic promise of this compound. Altogether, this demonstrates that marine natural product compounds have the potential to treat multidrug-resistant *A. baumannii* efflux pumps and support further investigations as ancillary therapeutic approaches.

## Supporting information

**S1 File. Excel file containing integrated datasets for *Acinetobacter baumannii* clinical isolates, including isolate metadata (Sheet 1), WGS accession details (Sheet 2), antimicrobial resistance and virulence gene profiles (Sheet 3), top 100 down- and upregulated genes with Gene Ontology enrichment (Sheets 4–5), and molecular dynamics analyses of the efflux pump: RMSD (Sheet 6), RMSF (Sheet 7), Rg (Sheet 8), and SASA (Sheet 9).**
(XLSX)

## Acknowledgments

The authors would like to take this opportunity to thank the management of Vellore Institute of Technology (VIT), Vellore, India, for providing the necessary facilities and encouragement to carry out this work.

## Author contributions

**Conceptualization:** Mohanraj Gopikrishnan, George Priya Doss C.

**Data curation:** Mohanraj Gopikrishnan.

**Formal analysis:** Mohanraj Gopikrishnan.

**Investigation:** Mohanraj Gopikrishnan.

**Methodology:** Mohanraj Gopikrishnan, George Priya Doss C.

**Project administration:** George Priya Doss C.

**Supervision:** George Priya Doss C.

**Visualization:** Mohanraj Gopikrishnan, George Priya Doss C.

**Writing – original draft:** Mohanraj Gopikrishnan.

**Writing – review & editing:** George Priya Doss C.

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
