## [Decision Letter · Decision Letter 0]

11 Sep 2025

Dear Dr. C Doss,

Thank you for submitting your manuscript to PLOS ONE. After careful consideration, we feel that it has merit but does not fully meet PLOS ONE’s publication criteria as it currently stands. Therefore, we invite you to submit a revised version of the manuscript that addresses the points raised during the review process.

We look forward to receiving your revised manuscript.

Kind regards,

Feng Gao

Academic Editor

PLOS ONE

Journal Requirements:

Mohanraj Gopikrishnan expresses gratitude to the Indian Council of Medical Research, India, for providing a Junior Research Fellowship (OMI/11/2020-ECD-1). The authors would like to take this opportunity to thank the management of Vellore Institute of Technology, Vellore, India, for providing the necessary facilities and encouragement to carry out this work.

5. Please note that funding information should not appear in any section or other areas of your manuscript. We will only publish funding information present in the Funding Statement section of the online submission form. Please remove any funding-related text from the manuscript.

Reviewers' comments:

Reviewer's Responses to Questions

**Comments to the Author**

1. Is the manuscript technically sound, and do the data support the conclusions?

Reviewer #1: Partly

Reviewer #2: Partly

Reviewer #3: Partly

2. Has the statistical analysis been performed appropriately and rigorously?

Reviewer #1: Yes

Reviewer #2: N/A

Reviewer #3: I Don't Know

3. Have the authors made all data underlying the findings in their manuscript fully available?

Reviewer #1: Yes

Reviewer #2: Yes

Reviewer #3: Yes

4. Is the manuscript presented in an intelligible fashion and written in standard English?

Reviewer #1: Yes

Reviewer #2: Yes

Reviewer #3: No

Reviewer #1: This study offers a comprehensive analysis of the pangenome-resistome and transcriptomic response of Acinetobacter baumannii to sub-MIC levels of several key antibiotics. Additionally, a virtual screening of marine natural products targeting the efflux transporter was conducted and identified several potential efflux pump inhibitors, followed by molecular dynamics simulations. The identification of promising marine-derived inhibitors, especially CMNPD27284, highlights potential candidates for future drug development against efflux-mediated resistance.

Major concerns:

1. This study relies heavily on online database analyses, lacking experimental validation to support the accuracy of its conclusions.

2. It is recommended to validate the results in clinical samples.

3. Discussion of other resistance mechanisms and possible interplay with MacB would provide a more comprehensive understanding of A. baumannii multidrug resistance.

4. The use of independent external datasets to verify the findings is strongly encouraged to ensure the robustness and reproducibility of the results.

5.The manuscript would benefit from careful language editing to improve clarity, grammar, and overall readability.

Reviewer #2: This study investigates the pangenome, resistome, and transcriptomic responses of Acinetobacter baumannii to sub-MIC antibiotic exposure, with a focus on efflux pump-mediated resistance. It also evaluates potential marine-derived inhibitors of the MacB efflux transporter through in silico screening and molecular dynamics simulations. While the study addresses an important question in antimicrobial resistance and employs a comprehensive, multi-layered analytical approach, several aspects of the methodology and data presentation would benefit from clarification and additional detail. My specific comments and suggestions are outlined below:

1. The authors describe a standard RNA-seq workflow for multiple A. baumannii strains, with experiments performed in duplicate as indicated in Table 1. While the overall approach, including quality assessment with FastQC, trimming with Trimmomatic, alignment with Subread, and quantification with featureCounts, is appropriate, there are several points that require clarification. The total sample size per condition should be explicitly stated in the text to ensure transparency. Furthermore, there is a discrepancy between the methodology and Figure 1, where HISAT2 and DESeq2 are indicated, whereas the Methods section describes Subread-align, featureCounts, and edgeR, which may cause confusion. To improve reproducibility, the authors should also provide details on quality filtering parameters, software versions, annotation release, and replicate design.

2. The use of PubMLST for Multi-Locus Sequence Typing and Roary for pan-genome analysis is appropriate, and Phandango visualization provides an effective means to explore gene presence and absence patterns. However, the section would benefit from additional details regarding the genomes analyzed, including accession numbers, source information, and assembly quality. While the thresholds for defining core genes are specified (>95% amino acid identity and >99% presence), other parameters used in Roary should be clarified. Similarly, more information on how Phandango was used for visualization would enhance clarity and reproducibility.

3. The authors appropriately applied Abricate with the CARD and VirulenceFinder databases to identify antimicrobial resistance and virulence-associated genes. To ensure reproducibility, it would be helpful to specify the exact database versions used, provide precise criteria or parameters for calling gene presence, and include accession numbers or sources for the genome assemblies analyzed. These additions would allow readers to better interpret the results and replicate the analysis.

4. The authors appropriately used edgeR for differential expression analysis, including normalization with the TMM method and statistical testing via the quasi-likelihood framework. However, the number of biological replicates per condition should be explicitly stated, as this is essential for assessing statistical power and reliability. There is a minor but important point regarding the reported thresholds: the manuscript mentions (log2FC) rather than |log2FC|. Using the absolute value is important when applying a fold-change cutoff irrespective of direction, and the authors should clarify whether up- and down-regulated genes were treated separately or together. Additionally, the authors should indicate whether batch effects were assessed or corrected.

Reviewer #3: The ms shows interesting in silico analyses from RNA seq data in A. baumannii. The ms is in general articulates well the rationale for their investigation. However, authors failed to provide a clear message that supports their claims. For example, the results section is written as a list of the techniques used for the analyses rather than a coherent story. It ewould be much better if authors used the obtained data and explain to readers the expected results for each of the analyses under a grouping that makes sense. In the currebt format is diffcult to assess. This is also the case for figs. even after downloading the tiffs. Figure 7 is unreadable and the figure legends do not provide suffient guidance to properly assess the data.

**Do you want your identity to be public for this peer review?** For information about this choice, including consent withdrawal, please see our Privacy Policy

Reviewer #1: No

Reviewer #2: **Yes:** Anika Bushra Lamisa

Reviewer #3: No

---

## [Author Response · Author response to Decision Letter 1]

28 Oct 2025

23-10-2025

To

The Editor-in-Chief,

PLOS One

Sub: Regarding the submission of the Revised Manuscript (PONE-D-25-45771_R1)

Dear Editor,

Please find our revised manuscript entitled “Integrative Pan-Resistome and Transcriptomic Characterization Reveals Differential Gene Expression Signatures in Carbapenem-Resistant Acinetobacter baumannii: Insights into Efflux Pump Regulation and Therapeutic Targeting Strategies” for consideration in the PLOS One. As per editorial and reviewer instructions, we have replied point-to-point to the reviewers’ comments. We are attaching a separate detailed response letter in this regard. We have incorporated the changes in the revised manuscript, and all the changes have been highlighted in the manuscript. Please find below the detailed point-to-point response/rebuttal to the reviewer comments received.

……………………………………………………………………………………………………………………………………………….

A POINT TO POINT RESPONSE TO REVIEWER’S COMMENTS

Manuscript Number: PONE-D-25-45771_R1

Reviewer #1:

Comment: This study relies heavily on online database analyses, lacking experimental validation to support the accuracy of its conclusions.

Response: We acknowledge the reviewer’s concern. This study is primarily computational, leveraging publicly available, high-quality RNA-Seq datasets and genome assemblies to ensure reproducibility and broad applicability. To enhance reliability, all docking experiments were performed in triplicate, and molecular dynamics simulations were extended up to 1000 ns to confirm the stability of ligand–protein interactions. While experimental validation in vitro and in vivo would indeed strengthen the findings, such validation is beyond the current scope of this study. We have clearly highlighted this as a limitation in the revised manuscript and proposed that future studies will extend these computational insights into experimental models.

Comment: It is recommended to validate the results in clinical samples.

Response: We agree with the reviewer that clinical validation would provide additional translational relevance. However, given that this study is focused on computational pangenome-resistome and transcriptomic analyses, validation in clinical isolates was not feasible within the current framework. Nevertheless, we ensured robustness by analyzing multiple strains under different antibiotic exposures, and by validating molecular interactions through independent replicates of docking and extended molecular dynamics simulations. Future work will aim to extend these findings to clinical samples to further substantiate the translational potential of the identified efflux pump inhibitors.

Comment: Discussion of other resistance mechanisms and possible interplay with MacB would provide a more comprehensive understanding of A. baumannii multidrug resistance.

Response: We thank the reviewer for this insightful comment. In the revised discussion, we have expanded our coverage of resistance mechanisms beyond MacB. Specifically, we now highlight the roles of other efflux pump families (RND-type AdeABC, AdeIJK, and AdeFGH), β-lactamase enzymes (OXA-type carbapenemases and NDM), and outer membrane porin modifications, which act synergistically to enhance resistance in A. baumannii. We also discuss the potential interplay between these systems and MacB, noting that efflux pumps often function in concert with enzymatic degradation and reduced membrane permeability to confer multidrug resistance. This integrated perspective underscores that while MacB is a critical target, resistance in A. baumannii is multifactorial, and therapeutic strategies should consider the broader network of resistance determinants.

Comment: The use of independent external datasets to verify the findings is strongly encouraged to ensure the robustness and reproducibility of the results.

Response: We appreciate the reviewer’s suggestion. While the present study primarily utilized a single comprehensive GEO dataset encompassing multiple strains and antibiotic exposures, we ensured robustness by analyzing each strain independently with biological replicates, which minimizes dataset-specific bias. Additionally, our computational workflow incorporated standard best-practice pipelines (FastQC, Trimmomatic, Subread-align, featureCounts, and edgeR) with stringent quality controls to ensure reproducibility. We acknowledge that the inclusion of independent external datasets would further strengthen the conclusions; however, suitable RNA-Seq datasets under comparable experimental conditions for A. baumannii were not available in the public domain at the time of analysis. We have highlighted this as a limitation and proposed that future work should extend these findings by validating them against independent transcriptomic datasets and clinical isolates.

Comment: The manuscript would benefit from careful language editing to improve clarity, grammar, and overall readability.

Response: We sincerely thank the reviewer for this valuable suggestion. The entire manuscript has been carefully revised for language, grammar, and clarity. We have also sought professional editing assistance from Grammarly to improve readability and ensure that the scientific content is conveyed more effectively.

Reviewer #2:

Comment: The authors describe a standard RNA-seq workflow for multiple A. baumannii strains, with experiments performed in duplicate as indicated in Table 1. While the overall approach, including quality assessment with FastQC, trimming with Trimmomatic, alignment with Subread, and quantification with featureCounts, is appropriate, there are several points that require clarification. The total sample size per condition should be explicitly stated in the text to ensure transparency. Furthermore, there is a discrepancy between the methodology and Figure 1, where HISAT2 and DESeq2 are indicated, whereas the Methods section describes Subread-align, featureCounts, and edgeR, which may cause confusion. To improve reproducibility, the authors should also provide details on quality filtering parameters, software versions, annotation release, and replicate design.

Response: We thank the reviewer for this valuable observation. The revised manuscript now explicitly states that 32 RNA-Seq datasets were analyzed, corresponding to four A. baumannii strains, each treated in duplicate with ciprofloxacin, amikacin, meropenem, and polymyxin B. We have also included software versions (FastQC v0.11.9, Trimmomatic v0.39, Subread-align v2.0.3, and featureCounts v2.0.3), filtering parameters (sliding window 4:20, minimum read length 36 bp), and the annotation source (NCBI RefSeq release 220, GTF format) to enhance reproducibility. In addition, Figure 1 has been corrected to accurately reflect the workflow using Subread-align, featureCounts, and edgeR instead of HISAT2 and DESeq2. These clarifications address the concerns raised and improve the transparency of our methodology.

Comment: The use of PubMLST for Multi-Locus Sequence Typing and Roary for pan-genome analysis is appropriate, and Phandango visualization provides an effective means to explore gene presence and absence patterns. However, the section would benefit from additional details regarding the genomes analyzed, including accession numbers, source information, and assembly quality. While the thresholds for defining core genes are specified (>95% amino acid identity and >99% presence), other parameters used in Roary should be clarified. Similarly, more information on how Phandango was used for visualization would enhance clarity and reproducibility.

Response: We thank the reviewer for these constructive comments. We have now included accession numbers, source information, and assembly quality metrics for the A. baumannii genomes analyzed. Additionally, we have clarified the parameters used in the Roary pipeline and expanded the description of how Phandango was applied for visualization purposes. These additions improve clarity and reproducibility. Section 2.2 has been revised accordingly.

Comment: The authors appropriately applied Abricate with the CARD and VirulenceFinder databases to identify antimicrobial resistance and virulence-associated genes. To ensure reproducibility, it would be helpful to specify the exact database versions used, provide precise criteria or parameters for calling gene presence, and include accession numbers or sources for the genome assemblies analyzed. These additions would enable readers to more effectively interpret the results and replicate the analysis.

Response: We appreciate the reviewer's valuable suggestion. We have now specified the exact versions of the CARD and VirulenceFinder databases used in our analysis, detailed the parameters applied for gene presence calling, and included accession numbers for all genome assemblies analyzed. These additions will enhance reproducibility, enabling readers to more accurately interpret and replicate our findings. Section 2.3 has been revised accordingly.

Comment: The authors appropriately used edgeR for differential expression analysis, including normalization with the TMM method and statistical testing via the quasi-likelihood framework. However, the number of biological replicates per condition should be explicitly stated, as this is essential for assessing statistical power and reliability. There is a minor but important point regarding the reported thresholds: the manuscript mentions (log2FC) rather than |log2FC|. Using the absolute value is important when applying a fold-change cutoff irrespective of direction, and the authors should clarify whether up- and down-regulated genes were treated separately or together. Additionally, the authors should indicate whether batch effects were assessed or corrected.

Response: We thank the reviewer for this thoughtful and constructive comment. We have now explicitly stated the number of biological replicates per condition in the Methods section to clarify the statistical power of our analysis. Regarding the differential expression thresholds, we confirm that we applied an absolute log₂ fold change cutoff (|log₂FC| ≥ 1), considering both up- and down-regulated genes. This has been corrected and clarified in the revised text. Additionally, we have carefully evaluated batch effects using principal component analysis (PCA), and no significant batch-related variation was observed across samples. This information has been added to the revised Methods section to enhance transparency and reproducibility.

Reviewer #3:

Comment: For example, the results section is written as a list of the techniques used for the analyses rather than a coherent story. It would be much better if authors used the obtained data and explained to readers the expected results for each of the analyses under a grouping that makes sense. In the current format is difficult to assess.

Response: We appreciate the reviewer's suggestion. In the revised manuscript, we have restructured the results section to present a coherent narrative, integrating the transcriptomic, genomic, and in silico analyses under thematic groupings rather than as a list of methods. Each subsection now begins with the key findings, followed by an interpretation that highlights how the data contribute to understanding A. baumannii’s antibiotic response and potential efflux pump inhibition.

Comment: This is also the case for figs. even after downloading the tiffs. Figure 7 is unreadable, and the figure legends do not provide sufficient guidance to properly assess the data.

Response: We appreciate the reviewer's attention to this matter. We acknowledge that the current version of Figure 7 lacks sufficient clarity. We have now revised Figure 7 to include a higher-resolution image, ensuring improved readability. Additionally, we have enhanced the figure legend to provide more detailed guidance for interpretation, thereby making the data presentation more transparent and more accessible.

---

## [Decision Letter · Decision Letter 1]

18 Dec 2025

Integrative Pan-Resistome and Transcriptomic Characterization Reveals Differential Gene Expression Signatures in Carbapenem-Resistant Acinetobacter baumannii: Insights into Efflux Pump Regulation and Therapeutic Targeting Strategies

PONE-D-25-45771R1

Dear Dr. C Doss,

We’re pleased to inform you that your manuscript has been judged scientifically suitable for publication and will be formally accepted for publication once it meets all outstanding technical requirements.

Kind regards,

Feng Gao

Academic Editor

PLOS One

Additional Editor Comments (optional):

Reviewers' comments:

Reviewer's Responses to Questions

**Comments to the Author**

Reviewer #1: All comments have been addressed

Reviewer #2: All comments have been addressed

2. Is the manuscript technically sound, and do the data support the conclusions?

Reviewer #1: Yes

Reviewer #2: Partly

3. Has the statistical analysis been performed appropriately and rigorously?

Reviewer #1: Yes

Reviewer #2: N/A

4. Have the authors made all data underlying the findings in their manuscript fully available?

Reviewer #1: Yes

Reviewer #2: Yes

5. Is the manuscript presented in an intelligible fashion and written in standard English?

Reviewer #1: Yes

Reviewer #2: Yes

Reviewer #1: (No Response)

Reviewer #2: The authors have addressed all comments raised in the previous review round, and the manuscript has improved substantially. The overall structure is clear, the study rationale is well presented, and the flow of the manuscript is easy to follow. The language is generally clear and written in standard English.

From a technical standpoint, the manuscript is partly sound, but a few areas still require clarification to fully support the conclusions. While the data provided appear consistent with the main findings, the authors should ensure that all methodological details—particularly those related to analytical decisions and assumptions—are described with sufficient depth to allow complete reproducibility. Although the manuscript does not include formal statistical analyses (as noted), any interpretation of quantitative results would benefit from additional explanation of uncertainty, limitations, and potential sources of bias.

The authors have complied with PLOS ONE’s data availability requirements, and all underlying data are appropriately provided or linked through repositories.

Overall, the manuscript is close to being suitable for publication, but the authors should consider strengthening the methodological explanations and clarifying any remaining ambiguities to ensure that readers can fully evaluate the robustness of the study.

**Do you want your identity to be public for this peer review?** For information about this choice, including consent withdrawal, please see our Privacy Policy

Reviewer #1: No

Reviewer #2: **Yes:** Anika Bushra Lamisa

---

## [Editor Report · Acceptance letter]

PONE-D-25-45771R1

PLOS One

Dear Dr. Doss C,

I'm pleased to inform you that your manuscript has been deemed suitable for publication in PLOS One. Congratulations! Your manuscript is now being handed over to our production team.

Kind regards,

on behalf of

Dr. Feng Gao

Academic Editor

PLOS One